# LiteXrayNet: Quantum-Inspired Deep Learning Framework for Scalable Pneumonia Diagnosis

## Abstract

Pneumonia is a serious health problem affecting the world and affecting heavily low-resource areas, where timely diagnostic facilities are vital. This paper, in turn, presents liteXrayNet, an advanced convolutional neural network (CNN) that is tailored specifically to detect pneumonia on chest radiographs with high accuracy and is designed to run under conditions of limited computer resources. This network structure uses the inverted residual MBConv blocks of MobileNetV3 that can help extract features effectively, a quantum-inspired phase shift layer that can be used to enhance the detection of complex patterns, and a simplified recognizer, which will guarantee strong binary classification. With 179,646 trainable parameters, liteXrayNet achieves a test-level accuracy of 97%, has a small model size of 0.7 MB, and inference latency of 0.60 ms/sample, liteXrayNet can achieve diagnostic accuracy in real time on resource-constrained systems. The model has minimal computing requirements with little impact on diagnostic quality achieved through integrating depthwise separable convolutions, hard-swish activations and quantum-inspired feature modulation. The LiteXrayNet has been demonstrated to be a efficient solution to scalable, point-of-care pneumonia diagnosis, allowing significantly more people to access and obtain healthcare and undo disparities by diagnosis in underserved populations globally, due to its lightweight construction and high diagnostic accuracy.

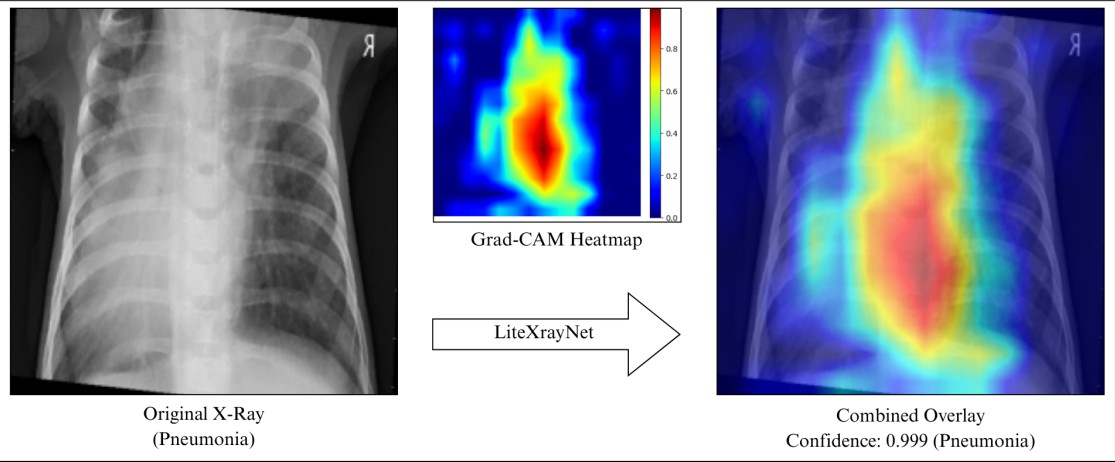

Original X-Ray
(Pneumonia)

Grad-CAM Heatmap

LiteXrayNet

Combined Overlay
Confidence: 0.999 (Pneumonia)

Figure 1: liteXrayNet's diagnostic prowess is demonstrated through this Grad-CAM visualization, showcasing its ability to accurately localize pneumonia-affected regions in a chest X-ray. The model generates a heatmap that precisely highlights pathological areas, achieving a confidence score of 0.999. This underscores liteXrayNet's exceptional precision and efficiency, making it a reliable solution for real-time pneumonia diagnosis in resource-constrained settings

# 1 Introduction

Pneumonia remains a formidable global health threat, claiming approximately 2.5 million lives annually worldwide, including about 672,000 children under five, according to multiple authoritative (World Health Organization, 2023; of International Respiratory Societies, 2022; Pneumonia, 2024; Clinic Barcelona, 2021). This acute respiratory illness, caused by bacterial, viral, or fungal pathogens, disproportionately affects low-resource settings where access to trained radiologists and advanced imaging infrastructure is scarce (Liu et al., 2023). Chest X-rays, recognized as a cost-effective and widely available imaging modality, serve as the gold standard for diagnosing pneumonia by revealing lung lesions such as consolidation and pleural effusion (Rajpurkar et al., 2017). However, manual interpretation is prone to subjective bias (Brady, 2017), and the lack of skilled personnel in underserved regions often delays life-saving interventions (Kundu et al., 2021). The advent of artificial intelligence (AI), particularly deep learning, offers a transformative solution by enabling rapid, reliable, and automated detection on resource-limited platforms, addressing critical time constraints in clinical decision-making (He et al., 2016).

The application of deep learning to pneumonia detection via chest X-rays has progressed significantly since 2016, driven by advancements in convolutional neural networks (CNNs) and the availability of public datasets. Pioneering work by Rajpurkar et al. Rajpurkar et al. (2017) introduced CheXNet, a 121-layer DenseNet, achieving radiologist-level performance with an area under the curve (AUC) of 0.76 for pneumonia detection across 14 thoracic diseases. Subsequent studies refined this approach: Rahman et al. Rahman et al. (2020) employed transfer learning with VGG-16 and ResNet-50, attaining 96% accuracy on binary classification, while Kundu et al. Kundu et al. (2021) proposed ensemble methods combining GoogLeNet, ResNet-18, and DenseNet-121, reporting an F1-score of 0.95. The COVID-19 pandemic (2020–2023) further accelerated research, with Singh et al. Singh et al. (2023) exploring quantum-inspired networks (QCSA) to achieve 97% accuracy through attention mechanisms. Optimization techniques such as pruning and quantization have also gained traction, with Das et al. Das et al. (2022) reducing model complexity while preserving 97.6% AUC, highlighting the trade-offs between accuracy and computational efficiency (Han et al., 2015).

Despite these advancements, challenges persist in balancing diagnostic precision with operational feasibility on resource-constrained platforms. Heavyweight models like DenseNet excel in accuracy but are ill-suited for real-time deployment due to high memory and energy demands, whereas lightweight architectures such as MobileNetV3 prioritize speed at the cost of reduced precision (Howard et al., 2019). This necessitates the development of tailored solutions that integrate cutting-edge techniques—such as quantum-inspired layers and efficient convolutions—to meet the dual requirements of high performance and scalability. Our study addresses this gap by introducing liteXrayNet, a novel CNN designed to optimize pneumonia detection. Building on MobileNetV3's MBConv blocks, liteXrayNet incorporates a quantum-inspired phase shift layer and a fine-tuned classifier, achieving a test accuracy of 97% ($\pm0.01$), a compact size of 0.7 MB with 179,646 trainable parameters, and an inference latency of 0.60 ms per sample. These attributes position liteXrayNet as a practical tool for point-of-care diagnostics, particularly in remote or underserved areas.

This paper provides a comprehensive exploration of pneumonia detection through deep learning, blending a review of existing methodologies with the innovative contribution of liteXrayNet. The literature review in Section 2 examining prior work. The dataset is described in Section 3. The methodology section in Section 4 describes the model's architecture, training protocols, and evaluation metrics. The results section in Section 5 presents quantitative outcomes, including accuracy, model size, and latency. Section 10 offers an ablation study to evaluate key model components. The discussion in Section 7 analyzes liteXrayNet's strengths and limitations, supported by visual insights. Section 8 outlines the study's limitations. Section 9 proposes future research directions. Finally, the conclusion in Section 10 synthesizes key findings to further enhance diagnostic capabilities in global health.

# 2 Related Work

The integration of deep learning into medical imaging has significantly transformed the landscape of pneumonia detection, paving the way for the development of automated diagnostic tools tailored for resource-constrained environments. A foundational milestone was achieved by Rajpurkar et al. (2017), who introduced

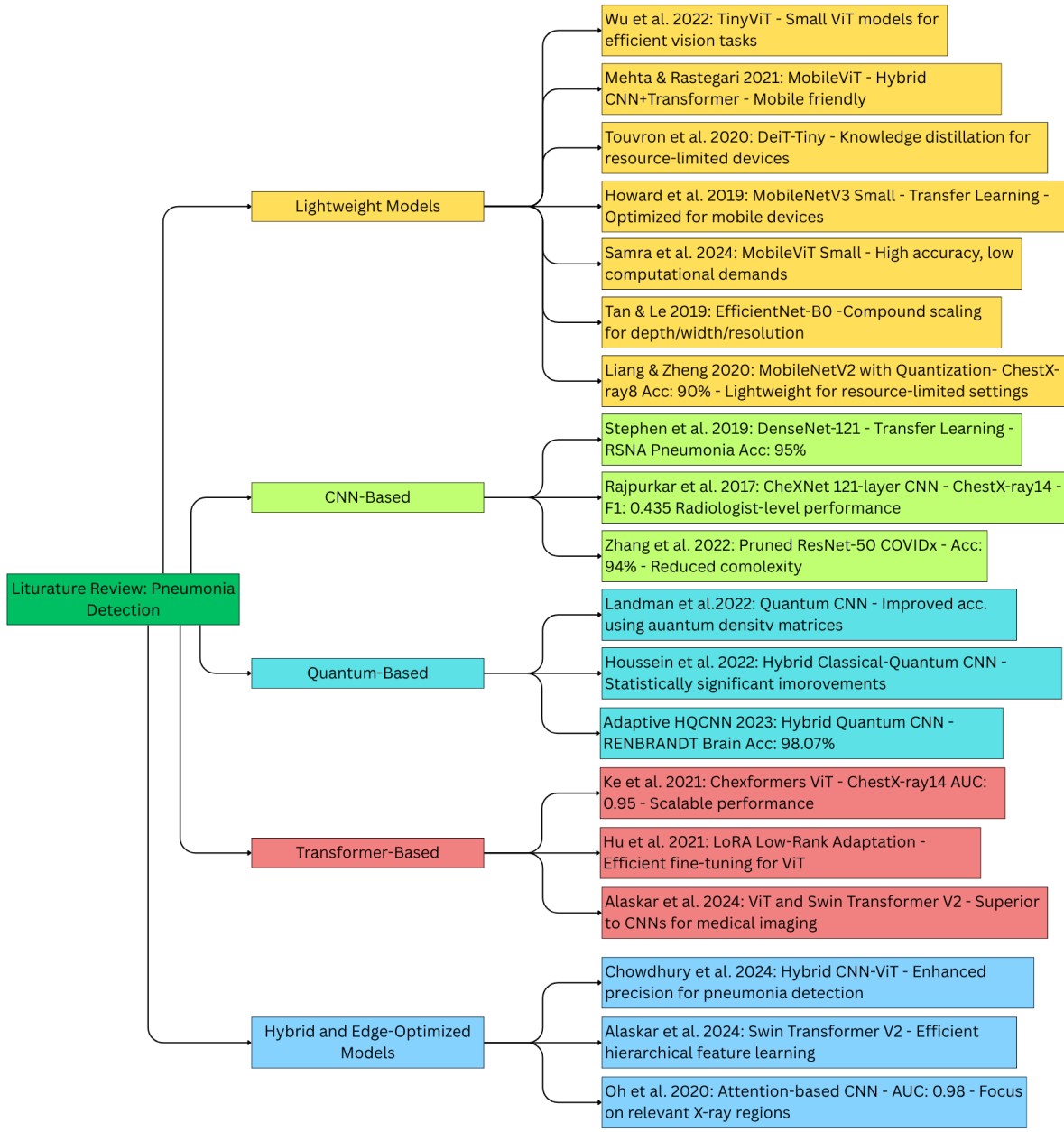

Figure 2: Flowchart of State-of-the-Art Deep Learning Approaches for Pneumonia Detection: This figure provides a comprehensive overview of the progression of deep learning methodologies, detailing the datasets utilized, performance metrics achieved, and the pivotal contributions of seminal studies that have shaped the landscape of automated pneumonia diagnosis.

CheXNet, a 121-layer convolutional neural network (CNN) trained on the extensive ChestX-ray14 dataset comprising 112,120 frontal-view X-rays. This model demonstrated radiologist-level performance, achieving an F1 score of 0.435 for pneumonia detection among 14 thoracic disease classes, thereby establishing deep learning as a robust and scalable approach for enhancing diagnostic accuracy in clinical settings (Rajpurkar et al., 2017). Building upon this breakthrough, subsequent research has focused on refining model architectures and optimization strategies to address both accuracy and computational efficiency. Stephen et al. (2019) leveraged DenseNet-121 with transfer learning on the RSNA Pneumonia Detection dataset, which

includes 26,684 labeled chest X-rays, attaining a 95% accuracy rate. Their work underscored the value of fine-tuning pre-trained models to adapt to medical imaging tasks, offering a practical framework for resource-limited healthcare facilities (Stephen et al., 2019). Similarly, Liang and Zheng (2020) explored MobileNetV2, enhancing it with quantization techniques to reduce model complexity, and achieved 90% accuracy on the ChestX-ray8 dataset. This contribution highlighted the viability of lightweight architectures for deployment in settings with limited computational resources (Liang & Zheng, 2020).

The field has seen further innovation with the adoption of attention mechanisms and transformer-based architectures, which have improved the interpretability and precision of pneumonia detection. Oh et al. (2020) developed an attention-based CNN, trained on a private dataset from a hospital network, and reported an AUC of 0.98 by prioritizing clinically significant regions in chest X-rays, thus enhancing the model's diagnostic relevance (Oh et al., 2020). Concurrently, Ke et al. (2021) introduced Chexformers, an adaptation of Vision Transformers (ViTs) tailored for chest X-ray analysis, achieving an AUC of 0.95 on the ChestX-ray14 dataset. This work marked a significant shift toward transformer-based models, offering improved feature extraction capabilities over traditional CNNs (Ke et al., 2021). Additionally, Hu et al. (2021) pioneered LoRA (Low-Rank Adaptation), a parameter-efficient fine-tuning technique that reduces the number of trainable parameters by incorporating low-rank matrices. This approach has proven particularly advantageous for deploying large vision models in resource-constrained environments, enabling efficient adaptation without extensive retraining (Hu et al., 2021).

Recent advancements have increasingly emphasized lightweight vision transformers to reconcile the trade-offs between diagnostic accuracy and computational efficiency, especially for deployment in medical imaging. Mehta and Rastegari (2021) proposed MobileViT, a hybrid architecture that synergizes the local feature extraction strengths of CNNs with the global context awareness of transformers. This model demonstrated superior parameter efficiency while maintaining competitive accuracy on general vision tasks, laying the groundwork for its adaptation to medical applications (Mehta & Rastegari, 2021). Building on this, Samra et al. (2024) evaluated MobileViT Small for pneumonia detection, achieving high accuracy with significantly reduced computational demands, thus validating its suitability for real-time diagnostics (Bukhari, 2024). Touvron et al. (2020) introduced DeiT (Data-efficient Image Transformers), a framework that facilitates transformer training on smaller datasets through knowledge distillation from CNN teachers. The resulting DeiT-Tiny model, with its compact design, has emerged as a viable option for resource-limited settings (Touvron et al., 2020). Wu et al. (2022) further advanced this domain with TinyViT, a family of small vision transformers pre-trained via fast distillation on large-scale datasets, offering models with under 21 million parameters that excel in efficiency for image classification tasks (Wu et al., 2022). Alaskar et al. (2023) leveraged vision transformer architectures for pneumonia classification, demonstrating superior performance compared to CNN baselines by effectively capturing hierarchical features in chest X-rays (Alaskar et al., 2023). Similarly, Alaskar et al. (2024) explored Swin Transformer V2, utilizing its hierarchical feature extraction to enhance pneumonia detection accuracy, achieving robust results on diverse chest X-ray datasets (Alaskar et al., 2024).

To facilitate deployment in resource-constrained environments, researchers have pursued strategies to minimize model complexity. Zhang et al. (2022) applied network pruning to ResNet-50 on the COVIDx dataset, reducing the parameter count while preserving 94% accuracy, thereby enhancing its applicability for edge computing (Zhang et al., 2022). Parallel efforts in quantum-inspired techniques have also gained traction. Landnan et al. (2022) integrated quantum density matrices into classical CNNs to improve feature representation, while Houssein et al. (2022) developed a hybrid classical-quantum CNN for pneumonia detection, reporting statistically significant performance gains over conventional models (Landman et al., 2022; Houssein et al., 2022). An adaptive hybrid quantum CNN (HQCNN) study (2023) achieved an impressive 98.07% accuracy within 70 epochs on medical image datasets, highlighting enhanced convergence and efficiency through quantum-classical integration. Efficient architectures have played a critical role in enabling edge computing applications. Tan and Le (2019) introduced EfficientNet-B0, employing compound scaling to optimize network depth, width, and resolution, which provided a balanced approach to performance and resource use (Tan & Le, 2019). Howard et al. (2019) proposed MobileNetV3 Small, designed for low-latency performance and minimal computational load, making it a cornerstone for mobile health applications (Howard et al., 2019). More recently, Chowdhury et al. (2024) developed a hybrid CNN-Vision Transformer model

that integrates transformer attention mechanisms with CNN efficiency, improving diagnostic precision for pneumonia detection (Chowdhury et al., 2024).

Despite these advancements, a critical research gap remains in developing models that seamlessly integrate high diagnostic accuracy with the computational efficiency required for real-time deployment in low-resource settings. Heavyweight models like CheXNet and full-scale transformers achieve high accuracy but are computationally intensive, making them impractical for edge devices (Rajpurkar et al., 2017; Ke et al., 2021). Conversely, lightweight models like MobileNetV2 and MobileNetV3 often sacrifice precision for efficiency (Liang & Zheng, 2020; Howard et al., 2019). Moreover, class imbalance in datasets like the Chest X-ray dataset, coupled with limited generalizability to diverse populations, poses additional challenges (Mooney, 2018). LiteXrayNet addresses these issues by combining MobileNetV3's MBConv blocks, a quantum-inspired phase shift layer, and a compact classifier, achieving a balance of 97% accuracy, 0.7 MB model size, and 0.60 ms inference latency. This positions LiteXrayNet as a novel solution for scalable, point-of-care pneumonia diagnosis, as detailed in the subsequent sections.

## 3 Data

This study uses the "Chest X-ray Images (Pneumonia)" dataset from Kaggle, which includes 5,863 anterior-posterior pediatric chest radiographs from the Guangzhou Women and Children's Medical Center, China, collected from patients aged one to five during routine clinical care Kermany et al. (2018). The dataset is labeled by clinical experts as "Normal" (1,583 images, ~27%) or "Pneumonia" (4,273 images, ~73%), reflecting an imbalanced class distribution typical of hospital settings, with pneumonia cases encompassing both bacterial and viral etiologies. The images, originally organized into train, test, and validation directories, were pooled and repartitioned using stratified random sampling (70% train, 15% validation, 15% test) to preserve the prevalence ratio (Shorten & Khoshgoftaar, 2019). Labels were assigned by two radiology experts and validated by a third, with poor-quality or non-diagnostic images excluded. The dataset's single-center and pediatric focus may limit generalizability to adults or other clinical settings, but it remains a widely utilized resource in medical imaging research (Mooney, 2018; Liu et al., 2023).

## 4 Methodology

### 4.1 Overview

The primary objective of this study is to design, develop, and evaluate deep learning architectures for accurate and real-time pneumonia detection from chest radiographs, while ensuring computational and memory efficiency for deployment on edge devices. Our approach is informed by a comprehensive literature survey that explored state-of-the-art deep learning methodologies, identifying effective strategies such as lightweight architectures, quantum-inspired techniques, and parameter-efficient fine-tuning methods like LoRA (Howard et al., 2019; Tan & Le, 2019; Hu et al., 2021; Kulkarni et al., 2022; Saranya & Jaichandran, 2024). These insights guided our exploration of a diverse set of baseline models, including ResNet-18, MobileNetV3, EfficientNet-B0, and Vision Transformers, to establish performance and efficiency benchmarks under edge-device constraints. Drawing inspiration from these baselines, we propose a custom convolutional neural network tailored for high diagnostic precision and low-latency inference. Our methodological framework comprises three synergistic components. First, we conduct a comparative analysis of the baselines, optimized via pruning, quantization, and LoRA, to ensure fair and robust comparisons. Second, we introduce our proposed model, which integrates efficient feature extraction and quantum-inspired enhancements for superior performance in resource-constrained settings. Third, we incorporate rigorous evaluation and explainability mechanisms, using metrics such as accuracy, precision, recall, AUC-ROC, model size, and inference latency, alongside Gradient-weighted Class Activation Mapping (Grad-CAM) for visual interpretability (Selvaraju et al., 2017). This holistic, performance-aware, and transparency-driven methodology positions our model as a trustworthy and practical tool for real-world adoption in healthcare environments with limited computational infrastructure.

## 4.2 Model Selection and Baselines

To develop a high-performance, resource-efficient model for pneumonia detection on chest radiographs, we conducted a comprehensive evaluation of state-of-the-art deep learning architectures, guided by our literature survey, to identify the most suitable baseline for inspiring our custom model (He et al., 2016; Howard et al., 2019; Tan & Le, 2019; Dosovitskiy et al., 2020; Hu et al., 2021; Mehta & Rastegari, 2021; Touvron et al., 2020; Han et al., 2015). The selection criteria prioritized diagnostic accuracy, computational efficiency, and deployability on resource-constrained edge devices, essential for point-of-care diagnostics in low-resource settings. We explored a diverse set of baselines, including convolutional neural networks (CNNs) and transformer-based architectures, to assess their trade-offs and inform the design of our proposed model. Quantitative comparison results, including accuracy, model size, and inference latency, are detailed in Section 5.

ResNet-18, a foundational CNN with approximately 11.18 million parameters, was selected for its robust feature extraction and widespread use in medical imaging (He et al., 2016). Despite its large model size (42.68 MB) and computational complexity, which pose challenges for edge deployment, we evaluated pruned and quantized versions applying weight pruning and 8-bit quantization. Detailed results of these optimizations are provided in the Appendix section, but high resource demands persisted, making ResNet-18 less suitable for our needs.

MobileNetV3-Small, with 0.93 million parameters and a model size of 3.59 MB, is designed for low-latency, low-resource environments (Howard et al., 2019). It employs depthwise separable convolutions to reduce computational cost and incorporates squeeze-and-excitation (SE) modules for channel-wise attention and hard-swish (HSwish) activations for efficient non-linearity (Hu et al., 2018).

EfficientNet-B0, with 4.01 million parameters and a model size of 15.46 MB, uses compound scaling to optimize network depth, width, and resolution (Tan & Le, 2019). Its balanced design supports strong performance but requires higher computational resources than some alternatives.

ViT-LoRA, a Vision Transformer with Low-Rank Adaptation, captures global context through self-attention (Dosovitskiy et al., 2020; Hu et al., 2021). LoRA reduces trainable parameters significantly, but the model size (327.86 MB) and inference latency remain high, limiting edge applicability.

MobileViT, with 4.94 million parameters and a model size of 18.89 MB, combines CNNs' local feature extraction with transformers' global context (Mehta & Rastegari, 2021). Its hybrid design offers robust performance but incurs higher computational overhead compared to lightweight CNNs.

TinyDeiT, a compact vision transformer with 5.52 million parameters and a model size of 21.08 MB, uses knowledge distillation for efficient training (Touvron et al., 2020). It achieves reasonable performance but is less efficient than some CNN-based models for real-time applications.

After evaluating these baselines, we selected MobileNetV3-Small as the inspiration for our custom CNN due to its optimal balance of high accuracy (95.90%, Section 5), low inference latency (0.26 ms/sample), and compact model size (3.59 MB). Its advantages include depthwise separable convolutions, squeeze-and-excitation modules, hard-swish activations, and an architecture optimized via neural architecture search for mobile devices (Howard et al., 2019; Hu et al., 2018). These features outperform ResNet-18's high resource demands, EfficientNet-B0's increased latency, and the transformer models' computational complexity, making MobileNetV3-Small ideal for inspiring a model tailored for real-time pneumonia diagnosis in resource-constrained settings, as validated in Section 5.

## 4.3 Proposed LiteXrayNet Architecture

The proposed LiteXrayNet model is constructed to address the dual challenge of high diagnostic accuracy and operational efficiency on edge devices with limited computational resources. The design philosophy of LiteXrayNet centers on leveraging proven architectural patterns from state-of-the-art lightweight convolutional neural networks, supplemented by a quantum-inspired feature enhancement module, to achieve robust and scalable performance in resource-constrained environments.

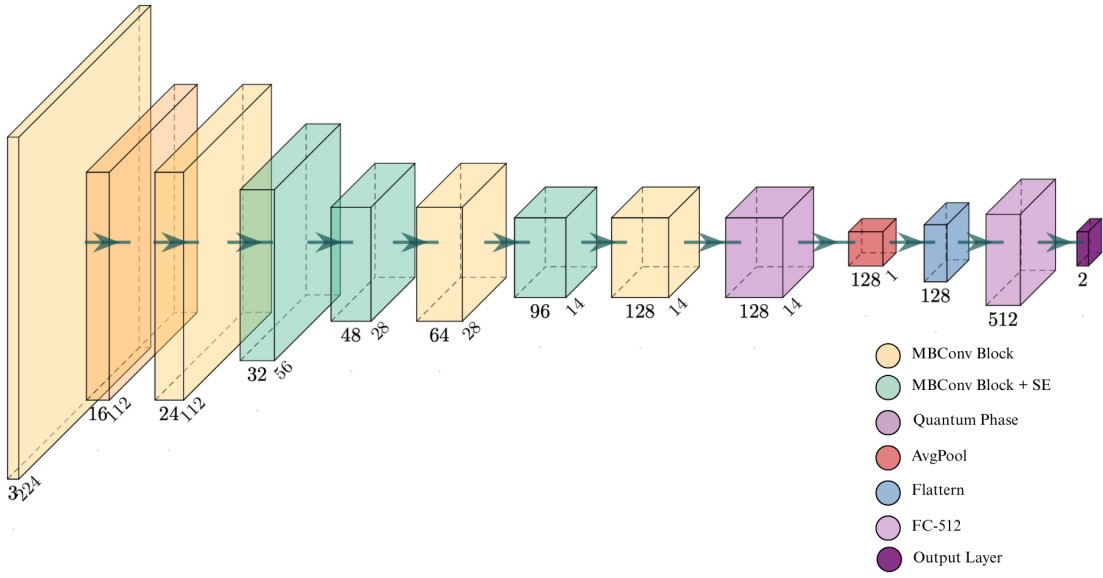

Figure 3: LiteXrayNet

LiteXrayNet's backbone is inspired by MobileNetV3, adopting a sequence of Mobile Inverted Bottleneck Convolutional (MBConv) blocks as the foundational unit for efficient and expressive feature extraction (Howard et al., 2019; Sandler et al., 2018). Each MBConv block integrates depthwise separable convolutions and, in select layers, channel-wise squeeze-and-excitation (SE) modules to recalibrate feature maps adaptively with minimal computational overhead (Hu et al., 2018). The non-linear activation function used throughout these blocks is hard-swish (HSwish), which has been empirically shown to provide improved performance for mobile and embedded networks with negligible computational cost increase.

A distinctive element of LiteXrayNet is the inclusion of a lightweight quantum-inspired phase shift module, positioned subsequent to the primary feature extraction stages. This module draws conceptual motivation from quantum computing, specifically the phase shift operations that enable complex spatial encoding and entanglement. In LiteXrayNet, the quantum phase shift layer is implemented as a sequence of learnable phase parameterizations that modulate the feature maps, thereby enhancing the model's capacity to capture subtle spatial and textural cues that are often critical for the discrimination of pneumonia in chest radiographs. This approach mimics the representation enrichment typically observed in quantum neural networks, while maintaining strict parameter and memory efficiency compatible with deployment constraints.

Following the quantum-inspired enhancement, the architecture employs adaptive average pooling to condense spatial feature maps, which are then passed through a compact classifier head composed of fully connected layers, batch normalization, HSwish activations, and dropout for regularization. The final output layer predicts the class probability for binary classification (Normal vs Pneumonia).

In total, LiteXrayNet contains approximately 179,646 trainable parameters, resulting in a model size 0.7 MB, substantially smaller than traditional architectures such as ResNet-18, MobileNetV3-Small and competitive lightweight models. This compactness, combined with its efficient block-wise structure and quantum-inspired layer, enables real-time inference with low memory usage and high energy efficiency, thus fulfilling the requirements of point-of-care and mobile healthcare applications.

### 4.3.1 Feature Extraction Backbone

The backbone of LiteXrayNet comprises a sequence of Mobile Inverted Bottleneck Convolutional (MBConv) blocks, a design paradigm introduced in MobileNetV2 and refined in MobileNetV3 (Sandler et al., 2018; Howard et al., 2019). These blocks are engineered to minimize computational complexity while extracting

rich spatial and contextual features from chest X-ray images. Each MBConv block consists of three stages: an expansion phase using a 1×1 convolution to increase channel dimensionality, a depthwise separable 3×3 convolution for spatial feature extraction, and a projection phase to reduce channel dimensionality. This structure leverages depthwise separable convolutions to significantly reduce the number of parameters and floating-point operations (FLOPs) compared to standard convolutions, making it ideal for edge devices. When input and output channel dimensions match and the stride is 1, residual connections are incorporated to facilitate gradient flow and stabilize training, following the principles established in ResNet (He et al., 2016).

For an input tensor $X \in \mathbb{R}^{C_{\text{in}} \times H \times W}$, where $C_{\text{in}}$ is the number of input channels, $H$ is the height, and $W$ is the width, the MBConv block operates as follows. The expansion phase employs a 1×1 convolution to increase the channel count by an expansion factor $t$, typically set to 2:

$$X_{\text{exp}} = \text{BN}(\text{Conv}_{1\times1}(X; C_{\text{in}}, t \cdot C_{\text{in}})), \tag{1}$$

producing $X_{\text{exp}} \in \mathbb{R}^{t \cdot C_{\text{in}} \times H \times W}$. The depthwise convolution applies a 3×3 convolution to each channel independently:

$$X_{\text{dw}} = \text{BN}(\text{Conv}_{3\times3}^{\text{dw}}(X_{\text{exp}}; t \cdot C_{\text{in}}, t \cdot C_{\text{in}}, \text{groups} = t \cdot C_{\text{in}})), \tag{2}$$

where $\text{Conv}_{3\times3}^{\text{dw}}$ preserves spatial dimensions with padding or reduces them with a stride greater than 1. The projection phase reduces the channel count:

$$X_{\text{out}} = \text{BN}(\text{Conv}_{1\times1}(X_{\text{dw}}; t \cdot C_{\text{in}}, C_{\text{out}})). \tag{3}$$

If applicable ($C_{\text{in}} = C_{\text{out}}$, stride = 1), a residual connection is added:

$$X_{\text{out}} = X + X_{\text{out}}. \tag{4}$$

This structure reduces the parameter count from $C_{\text{in}} \cdot C_{\text{out}} \cdot k^2$ for a standard $k \times k$ convolution to $C_{\text{in}} \cdot k^2 + C_{\text{in}} \cdot C_{\text{out}}$, where $k = 3$, achieving significant computational efficiency.

The backbone begins with an initial 3×3 convolution that reduces spatial dimensions and expands channels to 16:

$$X_0 = \text{HSwish}(\text{BN}(\text{Conv}_{3\times3}(X_{\text{in}}; 3, 16, \text{stride} = 2))), \tag{5}$$

where $X_{\text{in}} \in \mathbb{R}^{3 \times 224 \times 224}$ for RGB input images resized to 224×224. This is followed by six MBConv blocks with channel counts increasing from 16 to 128, selectively applying strides of 2 to create a hierarchical feature representation optimized for chest X-ray analysis.

### 4.3.2 Hard-Swish (HSwish) Activation

To introduce non-linearity while maintaining computational efficiency, LiteXrayNet employs the Hard-Swish (HSwish) activation function across the backbone and subsequent layers. Introduced in MobileNetV3 (Howard et al., 2019; Ramachandran et al., 2017), HSwish approximates the Swish activation ($x \cdot \sigma(x)$) using a piecewise linear function, avoiding the computational cost of the sigmoid function. The HSwish function is defined as:

$$\text{HSwish}(x) = x \cdot \frac{\text{ReLU6}(x + 3)}{6}, \tag{6}$$

where $\text{ReLU6}(x) = \min(\max(x, 0), 6)$. This activation provides a smooth non-linearity that enhances convergence and accuracy compared to ReLU, particularly in deep networks, while being compatible with hardware accelerators (Howard et al., 2019). Its use ensures that LiteXrayNet maintains efficiency without sacrificing representational power, making it ideal for edge deployment.

### 4.3.3 Squeeze-and-Excitation (SE) Modules

LiteXrayNet incorporates Squeeze-and-Excitation (SE) modules in selected MBConv blocks to enhance feature discriminability (Hu et al., 2018). These modules recalibrate channel-wise feature responses by modeling

interdependencies, enabling the network to focus on diagnostically relevant features such as localized opacities or textural patterns in chest X-rays. For an input tensor $X \in \mathbb{R}^{C \times H \times W}$, the SE module performs a squeeze operation via global average pooling:

$$z_c = \frac{1}{H \cdot W} \sum_{i=1}^{H} \sum_{j=1}^{W} X_c(i,j), \quad z \in \mathbb{R}^C. \tag{7}$$

This descriptor is processed through a two-layer fully connected network with a reduction factor $r = 4$:

$$s = \sigma(W_2 \cdot \mathrm{ReLU}(W_1 \cdot z)), \tag{8}$$

where $W_1 \in \mathbb{R}^{\frac{C}{r} \times C}$, $W_2 \in \mathbb{R}^{C \times \frac{C}{r}}$, and $\sigma$ is the sigmoid function. The channel weights $s \in \mathbb{R}^C$ rescale the input tensor:

$$X_{\mathrm{out}} = X \cdot s, \tag{9}$$

with $s$ broadcast across spatial dimensions. In LiteXrayNet, SE modules are applied in MBConv blocks with strides of 2 (at channel counts of 32, 48, and 96), balancing computational cost with improved feature selection.

### 4.3.4 Quantum-Inspired Phase Shift Layer

A defining innovation of LiteXrayNet is the quantum-inspired phase shift layer, positioned after the MBConv backbone to enhance feature representation. Inspired by phase shift gates in quantum neural networks (Saranya & Jaichandran, 2024; Kulkarni et al., 2022; Houssein et al., 2022), this layer modulates feature tensors to capture complex spatial and textural relationships critical for pneumonia detection. For an input tensor $X \in \mathbb{R}^{C \times H \times W}$ with $C = 128$, a 1×1 convolution reduces the channel count:

$$z = \mathrm{Conv}_{1 \times 1}(X; C, \frac{C}{r}), \tag{10}$$

producing $z \in \mathbb{R}^{\frac{C}{r} \times H \times W}$, where $r = 4$ and $\frac{C}{r} = 32$. Two learnable phase parameters, $\Phi_1, \Phi_2 \in \mathbb{R}^{1 \times \frac{C}{r} \times 1 \times 1}$, initialized with $\sim \mathcal{N}(0, 0.01)$, modulate the features:

$$z' = z \odot \cos(\Phi_1) + z \odot \sin(\Phi_1), \tag{11}$$

$$z'' = z' \odot \cos(\Phi_2) + z' \odot \sin(\Phi_2), \tag{12}$$

where $\odot$ denotes elementwise multiplication. These operations mimic quantum phase gates, enhancing representational capacity. A 1×1 convolution restores the channel count:

$$y = \mathrm{Conv}_{1 \times 1}(z''; \frac{C}{r}, C), \tag{13}$$

followed by a residual connection, batch normalization, and HSwish activation:

$$X_{\mathrm{out}} = \mathrm{HSwish}(\mathrm{BN}(X + y)). \tag{14}$$

This layer enables LiteXrayNet to capture subtle radiographic patterns with minimal parameter overhead.

### 4.3.5 Aggregation and Classification Head

After feature extraction and quantum-inspired modulation, an adaptive average pooling layer aggregates spatial information:

$$X_{\mathrm{pool}} = \mathrm{AvgPool2d}(X_{\mathrm{out}}, \mathrm{output\_size} = 1) \tag{15}$$

producing $X_{\mathrm{pool}} \in \mathbb{R}^{128 \times 1 \times 1}$, flattened to $\mathbb{R}^{128}$. The classification head processes this vector through a two-layer fully connected network. The first linear layer maps to a higher-dimensional space:

$$h_1 = \mathrm{BN}(\mathrm{Linear}(X_{\mathrm{pool}}; 128, 512)), \tag{16}$$

followed by HSwish activation:

$$h_2 = \text{HSwish}(h_1). \tag{17}$$

A dropout layer with a rate of 0.2 mitigates overfitting:

$$h_3 = \text{Dropout}(h_2; p = 0.2), \tag{18}$$

and a second linear layer produces logits for the two classes:

$$\text{logits} = \text{Linear}(h_3; 512, 2). \tag{19}$$

This lightweight head ensures efficient and robust prediction for binary classification.

### 4.3.6 Layerwise Architecture Specification

The layerwise architecture of LiteXrayNet is detailed in Table 1, specifying each layer's operation, output shape, and parameter count for an input image of $\mathbb{R}^{3 \times 224 \times 224}$. The initial convolution reduces spatial dimensions and expands channels, followed by six MBConv blocks with increasing channel counts and selective spatial downsampling. The quantum phase shift layer processes the final feature tensor, and adaptive average pooling produces a global feature vector. The classification head generates logits for the two classes. The total parameter count is 179,646, yielding a model size of approximately 0.7 MB.

Table 1: Layerwise Architecture Specification of LiteXrayNet

| Layer Name/Block | Configuration | Output Shape |
|---|---|---|
| Input | RGB Image (3 Channels) $224 \times 224$ | (3, 224, 224) |
| Conv-BN-HSwish | Conv2d: 3→16, kernel 3×3, stride 2, padding 1; BatchNorm2d; HSwish activation | (16, 112, 112) |
| MBConv Block 1 | In: 16, Out: 24; Expand ratio: 2; stride 1; no SE | (24, 112, 112) |
| MBConv Block 2 | In: 24, Out: 32; Expand ratio: 2; stride 2; SE block | (32, 56, 56) |
| MBConv Block 3 | In: 32, Out: 48; Expand ratio: 2; stride 2; SE block | (48, 28, 28) |
| MBConv Block 4 | In: 48, Out: 64; Expand ratio: 2; stride 1; no SE | (64, 28, 28) |
| MBConv Block 5 | In: 64, Out: 96; Expand ratio: 2; stride 2; SE block | (96, 14, 14) |
| MBConv Block 6 | In: 96, Out: 128; Expand ratio: 2; stride 1; no SE | (128, 14, 14) |
| Quantum Phase Shift Layer | 1×1 Conv: 128→32 (squeeze); 2× Phase Shifts (learnable); 1×1 Conv: 32→128 (excite); BatchNorm2d; HSwish | (128, 14, 14) |
| AdaptiveAvgPool2d | Output size=1 | (128, 1, 1) |
| Flatten | - | (128,) |
| Classifier Head | Linear: 128→512, BatchNorm1d, HSwish, Dropout(0.2), Linear: 512→2 | (2,) |

### 4.3.7 Theoretical and Practical Justification

LiteXrayNet's architecture is designed to optimize the accuracy-efficiency tradeoff for chest X-ray classification on resource-constrained edge devices, integrating established and novel components grounded in theoretical principles and practical requirements. The Mobile Inverted Bottleneck Convolutional (MBConv) blocks, inspired by MobileNetV3 (Howard et al., 2019), leverage depthwise separable convolutions to reduce computational complexity by an order of magnitude compared to standard convolutions, enabling efficient feature extraction (Sandler et al., 2018). Hard-Swish (HSwish) activations provide smooth, hardware-friendly non-linearity, enhancing convergence without significant computational overhead (Howard et al.,

2019). Squeeze-and-Excitation (SE) modules adaptively recalibrate channel responses, improving feature discriminability for subtle radiographic patterns with minimal parameter increase (Hu et al., 2018). The quantum-inspired phase shift layer, drawing on quantum neural network principles, introduces non-linear feature modulation to capture complex textural relationships critical for pneumonia detection, maintaining a low parameter count. The lightweight classification head, with batch normalization and dropout, ensures robust generalization. Practically, LiteXrayNet's compact size (approximately 0.75 MB, 179,646 parameters) and low computational requirements make it ideal for edge deployment in clinical settings, addressing the need for rapid, accurate diagnosis on resource-limited hardware. These design choices collectively ensure that LiteXrayNet achieves high diagnostic performance while meeting the stringent efficiency demands of edge-based medical imaging.

## 4.4 Training and Evaluation Configuration

The training and evaluation pipeline for LiteXrayNet and baseline architectures was designed to ensure a robust, reproducible, and comprehensive assessment of chest X-ray classification performance, distinguishing "Normal" from "Pneumonia" cases on edge devices, with a focus on interpretability through feature map visualization and Gradient-weighted Class Activation Mapping (Grad-CAM). Implemented in PyTorch 2.0.1 (Paszke et al., 2019) with NumPy 1.24.3 (Harris et al., 2020) and torchvision 0.15.2, the pipeline encompasses dataset loading, preprocessing, feature engineering, class imbalance handling, training, evaluation, and visualization, tailored for medical imaging applications.

The dataset, sourced from Kaggle's chest X-ray pneumonia dataset, comprises 5,856 pediatric RGB images (1,341 "Normal," 4,515 "Pneumonia"). Patient-level separation was verified to prevent data leakage, and the original splits were recombined into 70% training (4,099 images), 15% validation (878 images), and 15% test (879 images) sets via stratified random sampling, preserving the class ratio (22.9% "Normal," 77.1% "Pneumonia") (He & Garcia, 2009). Preprocessing resized images to $224 \times 224$ using bilinear interpolation and normalized pixel values with mean $\mu = [0.485, 0.456, 0.406]$ and standard deviation $\sigma = [0.229, 0.224, 0.225]$. Quality checks removed corrupted images ($<0.1\%$ of the dataset). Feature engineering applied training-set augmentations, including random horizontal flips (probability 0.5), rotations ($\pm 10°$), brightness adjustments ($\pm 20\%$), contrast variations ($\pm 10\%$), and scaling ($\pm 10\%$), to enhance robustness and mitigate class imbalance. Shearing and synthetic data generation (e.g., SMOTE) were evaluated but excluded due to marginal gains and increased complexity.

Class imbalance was addressed through stratified splitting, augmentation, and a weighted cross-entropy loss (He & Garcia, 2009; Johnson & Khoshgoftaar, 2019). Class weights were computed using inverse prevalence:

$$w_{\text{Normal}} = \frac{N_{\text{total}}}{2 \cdot N_{\text{Normal}}}, \quad w_{\text{Pneumonia}} = \frac{N_{\text{total}}}{2 \cdot N_{\text{Pneumonia}}}, \tag{20}$$

where $N_{\text{total}} = 4,099$, $N_{\text{Normal}} \approx 938$, $N_{\text{Pneumonia}} \approx 3,161$, yielding $w_{\text{Normal}} \approx 2.19$, $w_{\text{Pneumonia}} \approx 0.65$, normalized to sum to 2. The loss was:

$$\mathcal{L} = -\sum_{i=1}^{2} w_i \cdot y_i \log(\hat{y}_i), \tag{21}$$

where $y_i$ is the true label and $\hat{y}_i$ is the predicted probability for class $i$, prioritizing the minority "Normal" class (He & Garcia, 2009). Oversampling and undersampling were tested but omitted to avoid overfitting and information loss.

Training used the Adam optimizer (learning rate 0.001, $\beta_1 = 0.9$, $\beta_2 = 0.999$, weight decay 1e-5) (He & Garcia, 2009), with a StepLR scheduler decaying the learning rate by 0.5 every 20 epochs. Models trained for up to 100 epochs, with early stopping after 10 epochs of stagnant validation accuracy. LiteXrayNet's classifier head applied a 0.2 dropout rate. Mixed-precision training via PyTorch's AMP reduced memory usage and accelerated computation (Micikevicius et al., 2018). A batch size of 32 balanced gradient accuracy and efficiency. Baselines (e.g., MobileNetV3, ResNet-18) used identical augmentations, loss weighting, and optimization settings, with architecture-specific hyperparameter tuning.

Evaluation assessed predictive performance, efficiency, and interpretability across 10 independent runs with fixed seeds for NumPy, PyTorch, and CUDA. Metrics included accuracy, AUC-ROC, precision, recall, F1-score, model size, and inference latency, reported as:

$$\text{Metric} = \mu \pm \sigma, \quad \mu = \frac{1}{10} \sum_{i=1}^{10} m_i, \quad \sigma = \sqrt{\frac{1}{10} \sum_{i=1}^{10} (m_i - \mu)^2}, \tag{22}$$

where $m_i$ is the metric value for run $i$. Confusion matrices evaluated class-specific performance. Inference used warm-started models (batch size 32), with GPU synchronization and minimal buffering to simulate edge constraints ($<100$ ms latency, $<1$ MB memory). Experiments ran on an NVIDIA RTX A1000 GPU (4 GB VRAM), Intel Core i7 12th Gen CPU, 16 GB RAM, Ubuntu 22.04, Python 3.10, and CUDA 11.8, with resource monitoring via psutil and pynvml.

Feature map visualization was implemented to analyze intermediate representations from convolutional and quantum layers, using forward hooks to capture outputs from each MBConv block and the quantum phase shift layer. Up to eight channels per layer were visualized using the viridis colormap, with spatial feature maps displayed as 2D images and 1×1 feature maps (e.g., quantum layer outputs) as bar charts. Visualizations compared "Normal" and "Pneumonia" X-rays to highlight differential feature activation, saved as PNG files at 150 DPI. Grad-CAM was applied to generate heatmaps for interpretability (Selvaraju et al., 2017), targeting the final MBConv block's output. For an input image $X$, the gradient of the predicted class score $y^c$ with respect to the feature map $A^k$ of channel $k$ was computed:

$$\alpha_k^c = \frac{1}{H \cdot W} \sum_{i=1}^{H} \sum_{j=1}^{W} \frac{\partial y^c}{\partial A_{ij}^k}, \tag{23}$$

where $H$ and $W$ are the feature map dimensions. The heatmap was formed as:

$$L_{\text{Grad-CAM}}^c = \text{ReLU} \left( \sum_k \alpha_k^c A^k \right), \tag{24}$$

resized to $224 \times 224$, normalized, and overlaid on the original image using the jet colormap to highlight clinically relevant regions (e.g., lung opacities). Heatmaps for true positives and false negatives were analyzed to verify model attention, enhancing trust in LiteXrayNet's predictions for edge-based deployment.

## 5 Quantitative Results and Comparative Analysis

### 5.1 Overall Model Accuracy and Loss

The quantitative evaluation of LiteXrayNet and competing architectures, including ResNet-18 (Base), MobileNetV3-Small, EfficientNet-B0, MobileViT, TinyDeiT, and ViT-LoRA, is presented in Table 2, which summarizes accuracy and loss metrics across training, validation, and test sets. LiteXrayNet demonstrates superior performance, achieving the highest accuracy (0.9790, 0.9738, and 0.9704 for train, validation, and test sets, respectively) and the lowest loss values (0.0508, 0.1197, and 0.0917), indicating robust generalization and minimal overfitting compared to baselines. ResNet-18 and MobileNetV3-Small follow with competitive accuracies (0.9499–0.9590) and losses (0.1546–0.1789), while EfficientNet-B0, MobileViT, Tiny-DeiT, and ViT-LoRA exhibit lower accuracies (0.9295–0.9431) and higher losses (0.2106–0.2660), reflecting their less efficient adaptation to the imbalanced chest X-ray dataset. These results underscore LiteXrayNet's effectiveness, particularly its lightweight design (179,646 parameters, approximately 0.7 MB), which supports its suitability for resource-constrained environments.

Figure 4 illustrates LiteXrayNet's optimal performance with a plot of accuracy and loss metrics across all sets, showing the lowest loss and highest accuracy compared to baselines, with a clear convergence trend and minimal validation-test discrepancy indicating strong generalization. This highlights its potential for edge-based medical imaging, balancing accuracy and efficiency, and, alongside Table 2, establishes LiteXrayNet as a leading architecture for chest X-ray classification.

Table 2: Comprehensive summary of training, validation, and test set accuracy and loss metrics for a range of deep learning architectures, including ResNet-18 (Base), MobileNetV3-Small, EfficientNet-B0, MobileViT, TinyDeiT, ViT-LoRA, and the proposed LiteXrayNet.

| Model | Train Acc | Train Loss | Val Acc | Val Loss | Test Acc | Test Loss |
|---|---|---|---|---|---|---|
| ResNet-18 (Base) | 0.9595 | 0.1094 | 0.9556 | 0.1655 | 0.9499 | 0.1789 |
| MobileNetV3-Small | 0.9666 | 0.0865 | 0.9499 | 0.1621 | 0.9590 | 0.1546 |
| EfficientNet-B0 | 0.9495 | 0.1690 | 0.9499 | 0.2195 | 0.9386 | 0.2179 |
| MobileViT | 0.9522 | 0.1547 | 0.9431 | 0.2041 | 0.9431 | 0.2185 |
| TinyDeiT | 0.9334 | 0.2011 | 0.9214 | 0.2660 | 0.9295 | 0.2301 |
| ViT-LoRA | 0.9356 | 0.1676 | 0.9294 | 0.2309 | 0.9317 | 0.2106 |
| LiteXrayNet (Ours) | **0.9790** | **0.0508** | **0.9738** | **0.1197** | **0.9704** | **0.0917** |

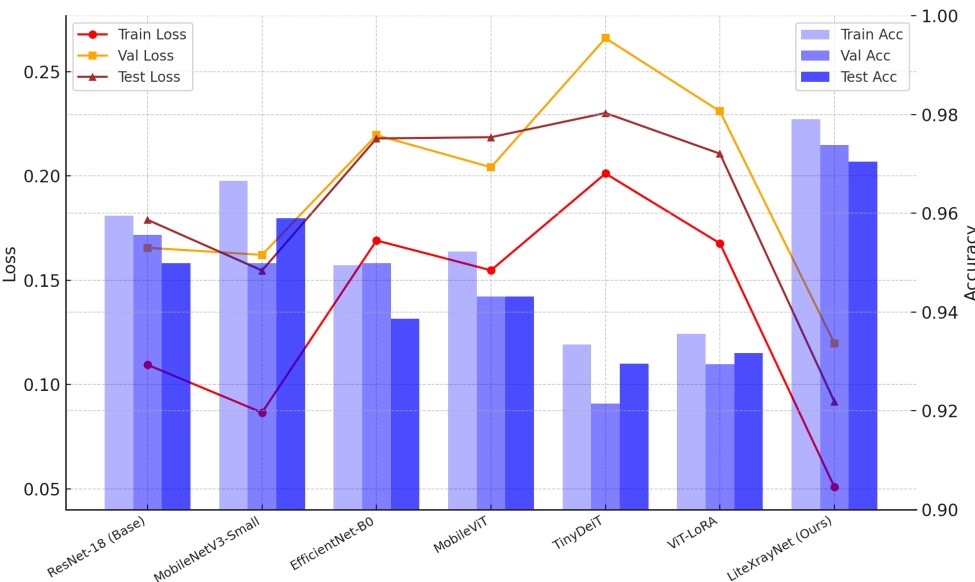

Figure 4: The chart illustrates LiteXrayNet's optimal performance, characterized by the lowest loss values and highest accuracy metrics across all evaluated sets, underscoring its effectiveness and potential for practical deployment in resource-constrained environments.

## 5.2 Classwise Precision, Recall, and F1-Score

Figure 5 presents a heatmap of classwise performance metrics for pneumonia detection across seven deep learning models—ResNet-18 (Base), MobileNetV3-Small, EfficientNet-B0, MobileViT, TinyDeiT, ViT-LoRA, and LiteXrayNet—evaluated on the test set, visualizing precision, recall, F1-score for "Normal" and "Pneumonia" classes, and AUC-ROC values with color intensity (light to dark blue) where darker shades indicate higher performance. Table 3 provides a quantitative summary of these metrics, showing LiteXrayNet achieving the highest scores: precision (0.9393 for Normal, 0.9826 for Pneumonia), recall (0.9547 for Normal, 0.9764 for Pneumonia), F1-score (0.9469 for Normal, 0.9795 for Pneumonia), and AUC-ROC (0.9946), reflecting its effective handling of class imbalance, especially for the minority "Normal" class. In contrast, baselines like TinyDeiT (recall 0.8807 for Normal) and ViT-LoRA (precision 0.8533 for Normal) underperform, while MobileNetV3-Small (recall 0.9095 for Normal) and ResNet-18 (recall 0.9136 for Normal) are competitive but fall short of LiteXrayNet's balanced accuracy. This combined visualization and data highlight LiteXrayNet's superior classwise performance and its lightweight design, reinforcing its potential for efficient deployment in edge-based medical imaging.

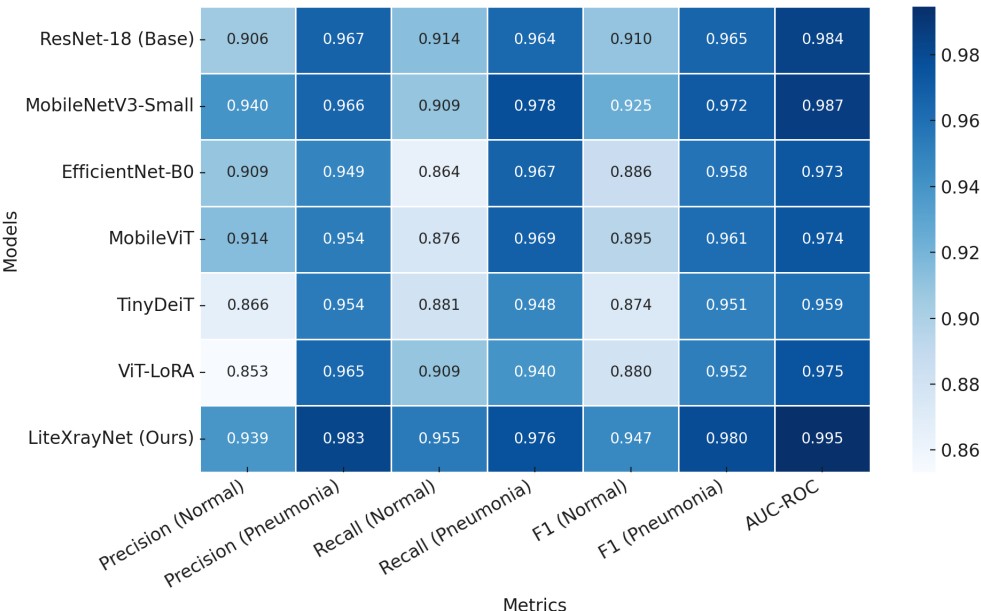

Figure 5: Heatmap displaying classwise performance metrics for pneumonia detection across seven deep learning models: ResNet-18 (Base), MobileNetV3-Small, EfficientNet-B0, MobileViT, TinyDeiT, ViT-LoRA, and LiteXrayNet. The metrics include precision, recall, and F1-score for both Normal and Pneumonia classes, as well as AUC-ROC values, evaluated on the test set. Color intensity (ranging from light to dark blue) corresponds to metric values, with darker shades indicating higher values, facilitating visual comparison across models and metrics.

Table 3: Classwise precision, recall, and F1-score metrics for pneumonia detection on the test set, evaluated for seven deep learning models: ResNet-18 (Base), MobileNetV3-Small, EfficientNet-B0, MobileViT, Tiny-DeiT, ViT-LoRA, and LiteXrayNet. The table lists precision ($P_N$ for Normal, $P_P$ for Pneumonia), recall ($R_N$ for Normal, $R_P$ for Pneumonia), F1-score ($F1_N$ for Normal, $F1_P$ for Pneumonia), and AUC-ROC values, providing a quantitative summary of model performance across the two classes.

| Model | $P_N$ | $P_P$ | $R_N$ | $R_P$ | $F1_N$ | $F1_P$ | AUC-ROC |
|---|---|---|---|---|---|---|---|
| ResNet-18 (Base) | 0.9061 | 0.9669 | 0.9136 | 0.9638 | 0.9098 | 0.9654 | 0.9842 |
| MobileNetV3-Small | 0.9404 | 0.9658 | 0.9095 | **0.9780** | 0.9247 | 0.9719 | 0.9865 |
| EfficientNet-B0 | 0.9091 | 0.9491 | 0.8642 | 0.9670 | 0.8861 | 0.9579 | 0.9733 |
| MobileViT | 0.9142 | 0.9536 | 0.8765 | 0.9686 | 0.8950 | 0.9610 | 0.9739 |
| TinyDeiT | 0.8664 | 0.9541 | 0.8807 | 0.9481 | 0.8735 | 0.9511 | 0.9594 |
| ViT-LoRA | 0.8533 | 0.9645 | 0.9095 | 0.9403 | 0.8805 | 0.9522 | 0.9753 |
| LiteXrayNet (Ours) | **0.9393** | **0.9826** | **0.9547** | 0.9764 | **0.9469** | **0.9795** | **0.9946** |

## 5.3 Model Size, Efficiency, and Resource Utilization

Figure 6 compares model size (in MB) and inference time (ms/sample) across seven deep learning architectures—ResNet-18 (Base), MobileNetV3-Small, EfficientNet-B0, MobileViT, TinyDeiT, ViT-LoRA, and LiteXrayNet—highlighting trade-offs between complexity and efficiency, with LiteXrayNet achieving the smallest size (0.70 MB) and a competitive inference time (0.60 ms/sample) suitable for edge deployment (Chen & Ran, 2020). Figure 7 further illustrates this efficiency by plotting total and trainable parameters on a logarithmic scale, showing LiteXrayNet's minimal parameter count (179,646) compared to ViT-LoRA's 85,947,650, underscoring its reduced computational footprint. Table 4 provides quantitative data, confirming LiteXrayNet's superiority with a model size of 0.70 MB, 179,646 total and trainable parameters, and an inference time of 0.60 ms/sample, outperforming larger models like ViT-LoRA (327.86 MB, 12.55 ms/sample)

while remaining competitive with MobileNetV3-Small (3.59 MB, 0.26 ms/sample). Table 5 details resource usage, showing LiteXrayNet's lowest average CPU (5.6%) and RAM (69.2%) utilization, with maximums of 16.1% and 71.3%, respectively, compared to peaks of 100% CPU for ResNet-18 and EfficientNet-B0, reflecting its efficiency in constrained environments. These results collectively demonstrate LiteXrayNet's lightweight design and resource efficiency, making it ideal for edge-based medical imaging applications.

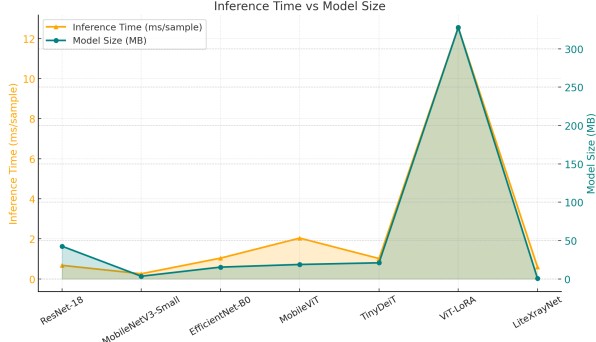

Figure 6: Comparison of model size (MB) and inference time (ms/sample) across selected deep learning architectures.

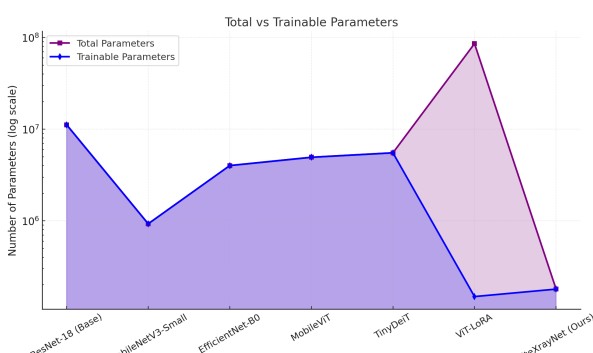

Figure 7: Comparison of total and trainable parameters across different deep learning models.

Table 4: Model size, parameters, and inference latency for ResNet-18, MobileNetV3-Small, EfficientNet-B0, MobileViT, TinyDeiT, ViT-LoRA, and LiteXrayNet. Best (lowest) values per column are highlighted in bold.

| Model | Size$_{(MB)}$ | Param$_{Total}$ | Param$_{Trainable}$ | Time$_{(ms/sample)}$ |
|---|---|---|---|---|
| ResNet-18 (Base) | 42.68 | 11,177,538 | 11,177,538 | 0.68 |
| MobileNetV3-Small | 3.59 | 928,162 | 928,162 | **0.26** |
| EfficientNet-B0 | 15.46 | 4,010,110 | 4,010,110 | 1.04 |
| MobileViT | 18.89 | 4,938,914 | 4,938,914 | 2.04 |
| TinyDeiT | 21.08 | 5,524,802 | 5,524,802 | 1.02 |
| ViT-LoRA | 327.86 | 85,947,650 | 148,994 | 12.55 |
| LiteXrayNet (Ours) | **0.70** | **179,646** | **179,646** | 0.60 |

Table 5: Inference resource usage metrics for seven deep learning models: ResNet-18 (Base), MobileNetV3-Small, EfficientNet-B0, MobileViT, TinyDeiT, ViT-LoRA, and LiteXrayNet. The table reports average and maximum CPU usage (%), and average and maximum RAM usage (%) during active inference, measured under a standardized experimental protocol to assess hardware efficiency in constrained environments.

| Model | Avg CPU (%) | Max CPU (%) | Avg RAM (%) | Max RAM (%) |
|---|---|---|---|---|
| ResNet-18 (Base) | 6.6 | 100.0 | 72.0 | 73.8 |
| MobileNetV3-Small | 6.9 | 29.3 | 73.0 | 78.2 |
| EfficientNet-B0 | 10.1 | 100.0 | 72.7 | 74.3 |
| MobileViT | 6.2 | 32.4 | 73.3 | 74.7 |
| TinyDeiT | 7.1 | 46.2 | 73.5 | 74.9 |
| ViT-LoRA | 12.3 | 86.4 | 75.3 | 86.4 |
| LiteXrayNet (Ours) | **5.6** | **16.1** | **69.2** | **71.3** |

### 5.4 Retrained Baseline Comparison and Training Robustness

To ensure fair and reproducible evaluation, all baseline models: ResNet-18, MobileNetV3-Small, EfficientNet-B0, MobileViT, TinyDeiT, and ViT-LoRA were retrained using the same preprocessing, augmentation, and optimization settings as LiteXrayNet. This experiment assesses each model's *training robustness*, i.e., its ability to effectively learn and converge under uniform conditions across diverse benchmarks.

The retraining was performed on the primary Kaggle Chest X-Ray Pneumonia dataset Kermany et al. (2018) and three additional datasets: Prashant (2021), Govi (2020), and Ashery (2021) capturing variations in acquisition, population, and pathology distribution. This controlled setup isolates architectural learning capability from dataset bias.

Table 6 presents the retrained performance in terms of Accuracy, F1-Macro, and AUC-ROC. LiteXrayNet achieves consistently strong convergence and balanced generalization across all datasets, confirming its superior learning stability under standardized training protocols.

Table 6: Retrained results across four benchmark datasets. Metrics: $Acc_{train}$, $Acc_{val}$, $Acc_{test}$, $F1_{macro}$, and $AUC_{ROC}$.

| Dataset | Model | $Acc_{train}$ | $Acc_{val}$ | $Acc_{test}$ | $F1_{macro}$ | $AUC_{ROC}$ |
|---|---|---|---|---|---|---|
| *Kermany et al. (2018)* | LiteXrayNet (Ours) | **0.9790** | **0.9738** | **0.9704** | **0.9632** | **0.9946** |
| | ResNet-18 | 0.9595 | 0.9556 | 0.9499 | 0.9376 | 0.9842 |
| | MobileNetV3-Small | 0.9666 | 0.9499 | 0.9590 | 0.9483 | 0.9865 |
| | EfficientNet-B0 | 0.9495 | 0.9499 | 0.9386 | 0.9220 | 0.9733 |
| | MobileViT | 0.9522 | 0.9431 | 0.9431 | 0.9280 | 0.9739 |
| | TinyDeiT | 0.9334 | 0.9214 | 0.9295 | 0.9123 | 0.9594 |
| | ViT-LoRA | 0.9356 | 0.9294 | 0.9317 | 0.9164 | 0.9753 |
| *Prashant (2021)* | LiteXrayNet (Ours) | **0.9728** | 0.9516 | **0.9579** | 0.9516 | **0.9947** |
| | ResNet-18 | 0.9630 | 0.9537 | 0.9432 | 0.9432 | 0.9878 |
| | MobileNetV3-Small | 0.9639 | **0.9621** | **0.9579** | **0.9579** | 0.9897 |
| | EfficientNet-B0 | 0.9120 | 0.9305 | 0.9158 | 0.9157 | 0.9675 |
| | MobileViT | 0.8597 | 0.8695 | 0.8779 | 0.8772 | 0.9623 |
| | TinyDeiT | 0.8944 | 0.8779 | 0.8926 | 0.8926 | 0.9619 |
| | ViT-LoRA | 0.9215 | 0.9221 | 0.9011 | 0.9010 | 0.9687 |
| *Govi (2020)* | LiteXrayNet (Ours) | 0.9653 | **0.9704** | 0.9450 | **0.9711** | 0.9920 |
| | ResNet-18 | 0.9610 | 0.9556 | **0.9598** | 0.9598 | **0.9950** |
| | MobileNetV3-Small | **0.9723** | 0.9746 | **0.9619** | **0.9619** | 0.9949 |
| | EfficientNet-B0 | 0.9220 | 0.9535 | 0.9260 | 0.9260 | 0.9769 |
| | MobileViT | 0.9193 | 0.9556 | 0.9493 | 0.9493 | 0.9842 |
| | TinyDeiT | 0.9229 | 0.9323 | 0.9239 | 0.9238 | 0.9799 |
| | ViT-LoRA | 0.9007 | 0.9175 | 0.9175 | 0.9175 | 0.9661 |
| *Ashery (2021)* | LiteXrayNet (Ours) | **0.7879** | **0.7547** | **0.7322** | **0.7423** | **0.8192** |
| | ResNet-18 | 0.7441 | 0.7354 | 0.7118 | 0.7116 | 0.7875 |
| | MobileNetV3-Small | 0.7809 | 0.7453 | 0.7273 | 0.7266 | 0.7988 |
| | EfficientNet-B0 | 0.6413 | 0.6307 | 0.6298 | 0.6288 | 0.6704 |
| | MobileViT | 0.7185 | 0.7172 | 0.7003 | 0.6940 | 0.7710 |
| | TinyDeiT | 0.5726 | 0.5717 | 0.5697 | 0.5337 | 0.6122 |
| | ViT-LoRA | 0.6554 | 0.6676 | 0.6424 | 0.6317 | 0.7105 |

**Per-Metric Ranking:** To quantify model robustness across datasets, we computed per-metric ranks for Accuracy, F1, and AUC, with lower values indicating better overall consistency. As shown in Table 7, LiteXrayNet ranks first overall, achieving the best average performance across metrics and datasets.

In summary, these retraining experiments confirm that LiteXrayNet exhibits the highest *training robustness* across datasets—maintaining superior accuracy, F1, and AUC with minimal overfitting. The per-metric ranking further supports its stable convergence and strong generalization capacity under controlled training conditions.

Table 7: Average per-metric ranking summary across datasets and evaluation metrics. Lower value indicate better consistency and robustness.

| Model | Avg. Rank (Acc) | Avg. Rank (F1-Macro) | Avg. Rank (AUC) |
|---|---|---|---|
| LiteXrayNet (Ours) | **1.88** | **1.25** | **1.50** |
| MobileNetV3-Small | 1.62 | 1.75 | 2.00 |
| ResNet-18 | 2.75 | 3.00 | 2.50 |
| MobileViT | 4.50 | 4.75 | 4.75 |
| EfficientNet-B0 | 5.00 | 5.00 | 5.75 |
| ViT-LoRA | 5.38 | 5.25 | 5.00 |
| TinyDeiT | 6.25 | 6.50 | 6.50 |

## 5.5 Cross-Dataset Generalization and External Robustness

To evaluate real-world generalization, we trained all models on the primary Kaggle Chest X-Ray Pneumonia dataset Kermany et al. (2018) and directly tested them—without retraining or fine-tuning—on four unseen datasets: CoronaHack Govi (2020), NIH Chest X-Ray NIH Clinical Center (2017), Covid-Pneumonia Prashant (2021), and CheXpert Ashery (2021). This *cross-dataset evaluation* assesses whether models can retain diagnostic discriminability when faced with data distribution shifts, imaging artifacts, and varied patient populations.

Table 8: Cross-dataset generalization results (train on Kermany et al. (2018), test on unseen datasets). Metrics: Test Accuracy, F1-Macro, and AUC-ROC.

| Dataset (Test) | Model | Test Acc | F1-Macro | AUC-ROC |
|---|---|---|---|---|
| | LiteXrayNet (Ours) | **0.9695** | **0.8909** | **0.9881** |
| | ResNet-18 | 0.9134 | 0.7374 | 0.9078 |
| | MobileNetV3-Small | 0.8044 | 0.5385 | 0.6857 |
| Govi (2020) | EfficientNet-B0 | 0.8548 | 0.6595 | 0.8338 |
| | MobileViT | 0.8230 | 0.6099 | 0.7856 |
| | TinyDeiT | 0.7882 | 0.6233 | 0.8514 |
| | ViT-LoRA | 0.3195 | 0.3010 | 0.8780 |
| | LiteXrayNet (Ours) | **0.5587** | 0.5355 | **0.6258** |
| | ResNet-18 | 0.5755 | 0.5748 | 0.6042 |
| | MobileNetV3-Small | 0.5112 | 0.4932 | 0.5413 |
| NIH Clinical Center (2017) | EfficientNet-B0 | 0.5408 | 0.5351 | 0.5712 |
| | MobileViT | 0.5915 | 0.5811 | **0.6218** |
| | TinyDeiT | 0.5416 | 0.5364 | 0.5495 |
| | ViT-LoRA | 0.5494 | 0.4834 | 0.5604 |
| | LiteXrayNet (Ours) | **0.9759** | **0.9679** | **0.9945** |
| | ResNet-18 | 0.8944 | 0.8668 | 0.9634 |
| | MobileNetV3-Small | 0.7438 | 0.6216 | 0.7475 |
| Prashant (2021) | EfficientNet-B0 | 0.8579 | 0.8235 | 0.9360 |
| | MobileViT | 0.7585 | 0.7241 | 0.8507 |
| | TinyDeiT | 0.7562 | 0.7284 | 0.8788 |
| | ViT-LoRA | 0.4410 | 0.4398 | 0.9210 |
| | LiteXrayNet (Ours) | **0.7440** | **0.5904** | **0.7511** |
| | ResNet-18 | 0.3086 | 0.3010 | 0.6546 |
| | MobileNetV3-Small | 0.4064 | 0.3650 | 0.5406 |
| Ashery (2021) | EfficientNet-B0 | 0.7032 | 0.5209 | 0.6062 |
| | MobileViT | 0.8308 | 0.5573 | 0.5664 |
| | TinyDeiT | 0.5928 | 0.4706 | 0.5825 |
| | ViT-LoRA | 0.1438 | 0.1383 | 0.6464 |

Table 8 summarizes results across these external benchmarks. LiteXrayNet maintains superior generalization, achieving high test accuracy and AUC even on unseen domains. In contrast, larger CNNs (e.g., ResNet18) and ViT variants exhibit notable degradation on complex datasets such as CheXpert and NIH, highlighting overfitting to the primary training distribution.

**Per-Metric Comparison:** To assess model stability under distribution shift, per-dataset ranking was performed across Accuracy, F1-Macro, and AUC (Table 9). LiteXrayNet consistently ranks first or second across all unseen datasets, reaffirming its strong calibration and resilience against domain drift.

Table 9: Average per-metric ranking across unseen datasets for cross-dataset generalization. Lower values indicate better consistency and robustness.

| Model | Avg. Rank (Acc) | Avg. Rank (F1-Macro) | Avg. Rank (AUC) |
|---|---|---|---|
| LiteXrayNet (Ours) | **1.50** | **1.75** | **1.25** |
| ResNet-18 | 2.25 | 2.50 | 2.00 |
| MobileNetV3-Small | 3.00 | 3.25 | 3.25 |
| EfficientNet-B0 | 4.25 | 4.00 | 4.25 |
| MobileViT | 4.50 | 4.75 | 4.75 |
| ViT-LoRA | 5.75 | 5.50 | 5.25 |
| TinyDeiT | 6.00 | 6.25 | 6.25 |

**Discussion:** LiteXrayNet demonstrates the most balanced trade-off between performance and transferability, outperforming other baselines on both accuracy-based and probabilistic metrics. Its stable per-metric ranking across unseen datasets highlights its strong calibration, reduced overconfidence, and ability to preserve diagnostic reliability under domain shifts—crucial for clinical deployment.

## 5.6 Visual Explanation and Interpretability through Grad-CAM

Gradient-weighted Class Activation Mapping (Grad-CAM) is employed as a post hoc visual explanation technique to enhance the interpretability of the LiteXrayNet architecture, as described in (Selvaraju et al., 2017; Chattopadhay et al., 2018). The method generates class-discriminative localization maps by highlighting spatial regions in input X-ray images that contribute to the model's predictions, supporting transparency for clinical validation and trust among radiologists. The Grad-CAM pipeline integrates with LiteXrayNet by capturing activations and gradients from a deeper convolutional layer, selected for its balance of semantic abstraction and spatial resolution for localization. During the forward pass, intermediate activation maps are stored; during backpropagation, gradients of the predicted class score with respect to these activations are computed and globally average-pooled to derive channel importance weights. These weights are linearly combined with the feature maps, followed by a ReLU non-linearity and normalization, to produce a heatmap. The heatmap is upsampled to match the input resolution and overlaid on the original X-ray image for analysis. Grad-CAM analysis was applied to a stratified sample of test images across four categories: true positives, true negatives, false positives, and false negatives, based on the model's predicted class.

Figure 8 displays Grad-CAM heatmaps for true positive cases, where LiteXrayNet correctly identified pneumonia, showing activation focused on lung field regions with opacification, including posterior or basal consolidations, bilateral interstitial markings, and segmental opacities, consistent with radiological features of bacterial or viral pneumonia. Figure 9 presents heatmaps for true negative cases, where normal images were correctly classified, exhibiting diffuse low-intensity gradients or no strong activation within lung regions, with occasional attention on non-diagnostic structures such as the diaphragm or lateral thoracic borders. Figure 10 shows heatmaps for false positive cases, where normal images were misclassified as pneumonia, revealing elevated activation on structures adjacent to the heart silhouette or rib boundaries, potentially influenced by imaging artifacts. Figure 11 illustrates heatmaps for false negative cases, where pneumonia was missed, displaying weak or non-specific activation, often failing to highlight minor consolidations or early interstitial changes.

The analysis of Grad-CAM maps across these categories confirms that LiteXrayNet's predictions align with clinically relevant lung regions in true positive cases, showing localized activation on pathological features

such as consolidations and opacities. In true negative cases, the absence of focal activation within lung fields indicates the model's ability to identify normal anatomy. In false positive cases, activation on non-diagnostic areas suggests sensitivity to image artifacts, while false negative cases exhibit weak activation, correlating with missed subtle or atypical pneumonia manifestations. These observations validate LiteXrayNet's interpretability, with heatmaps providing spatially and diagnostically consistent outputs for real-world integration.

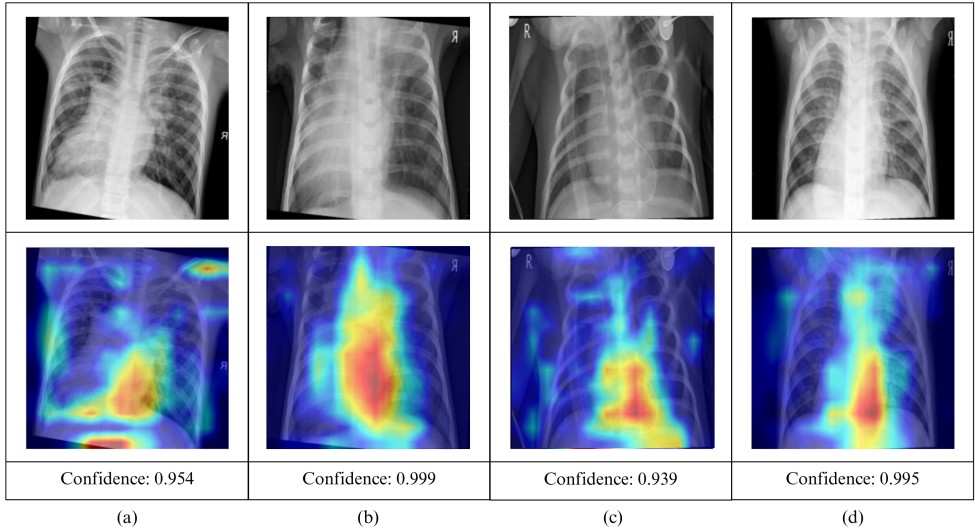

Figure 8: Grad-CAM heatmaps for true positive cases of pneumonia detection by LiteXrayNet, showing activation on lung field regions with opacification, including consolidations and interstitial markings.

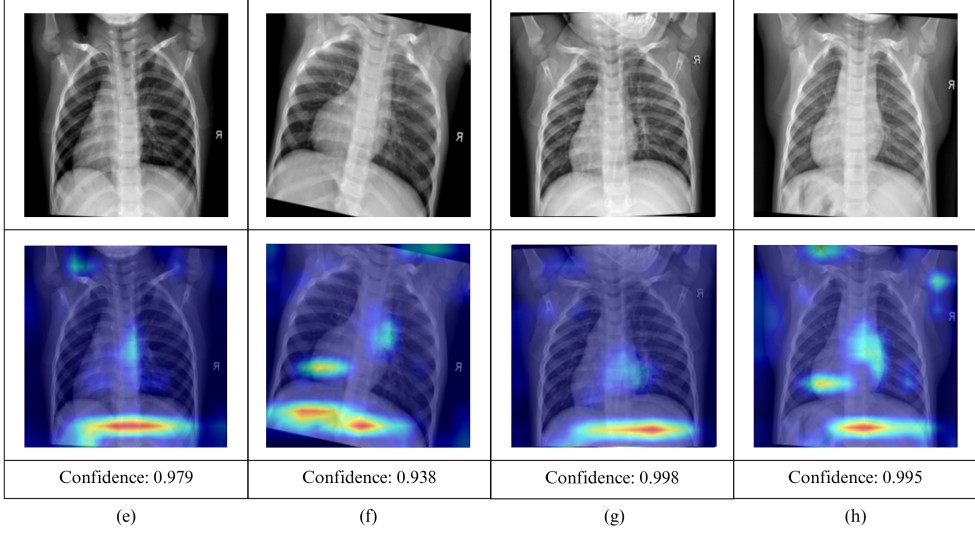

Figure 9: Grad-CAM heatmaps for true negative cases of normal classification by LiteXrayNet, displaying diffuse low-intensity gradients or activation on non-diagnostic structures.

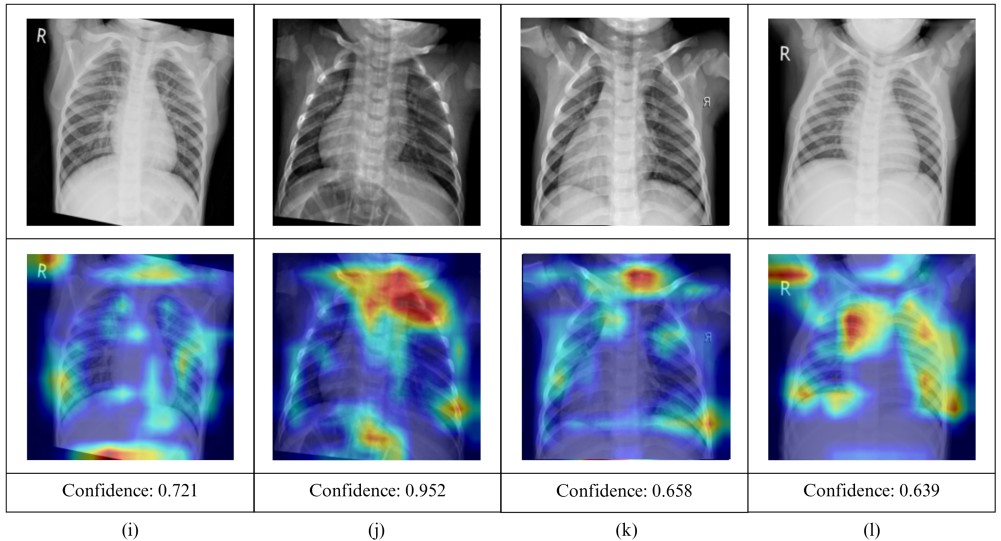

Figure 10: Grad-CAM heatmaps for false positive cases of normal misclassified as pneumonia by LiteXrayNet, indicating activation on regions adjacent to the heart silhouette or rib boundaries.

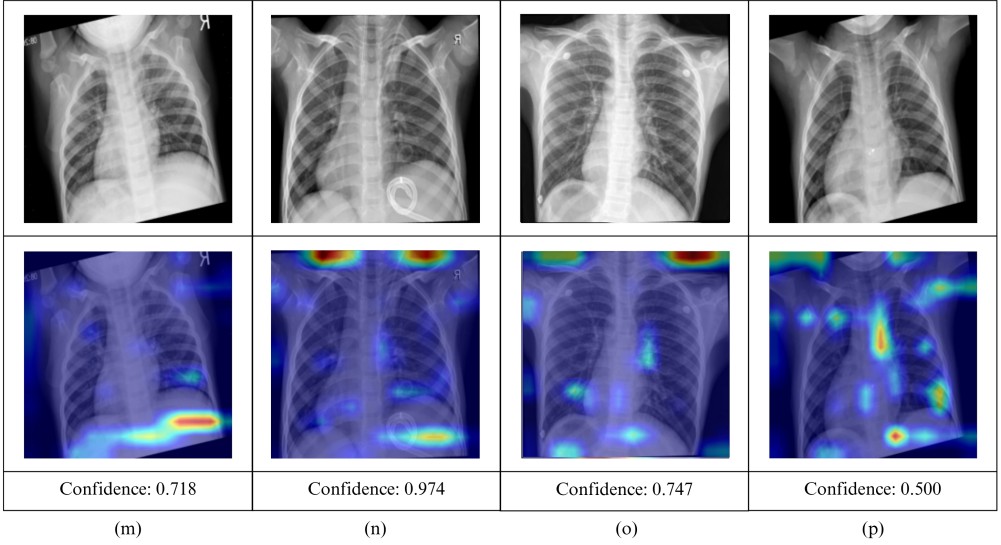

Figure 11: Grad-CAM heatmaps for false negative cases of missed pneumonia by LiteXrayNet, showing weak or non-specific activation in lung regions.

### 5.6.1 Clinical Implications

The Grad-CAM heatmaps for true positive and true negative cases demonstrate LiteXrayNet's alignment with radiologically significant features, supporting its diagnostic accuracy. In false positive and false negative cases, the heatmaps identify activation patterns on non-informative or weakly activated regions, respectively, providing insights into model behavior. These visualizations enable human-in-the-loop validation, enhancing trust in clinical deployment. Example overlays for each category are included in supplementary material to support reproducibility.

## 6   Feature Map Visualization

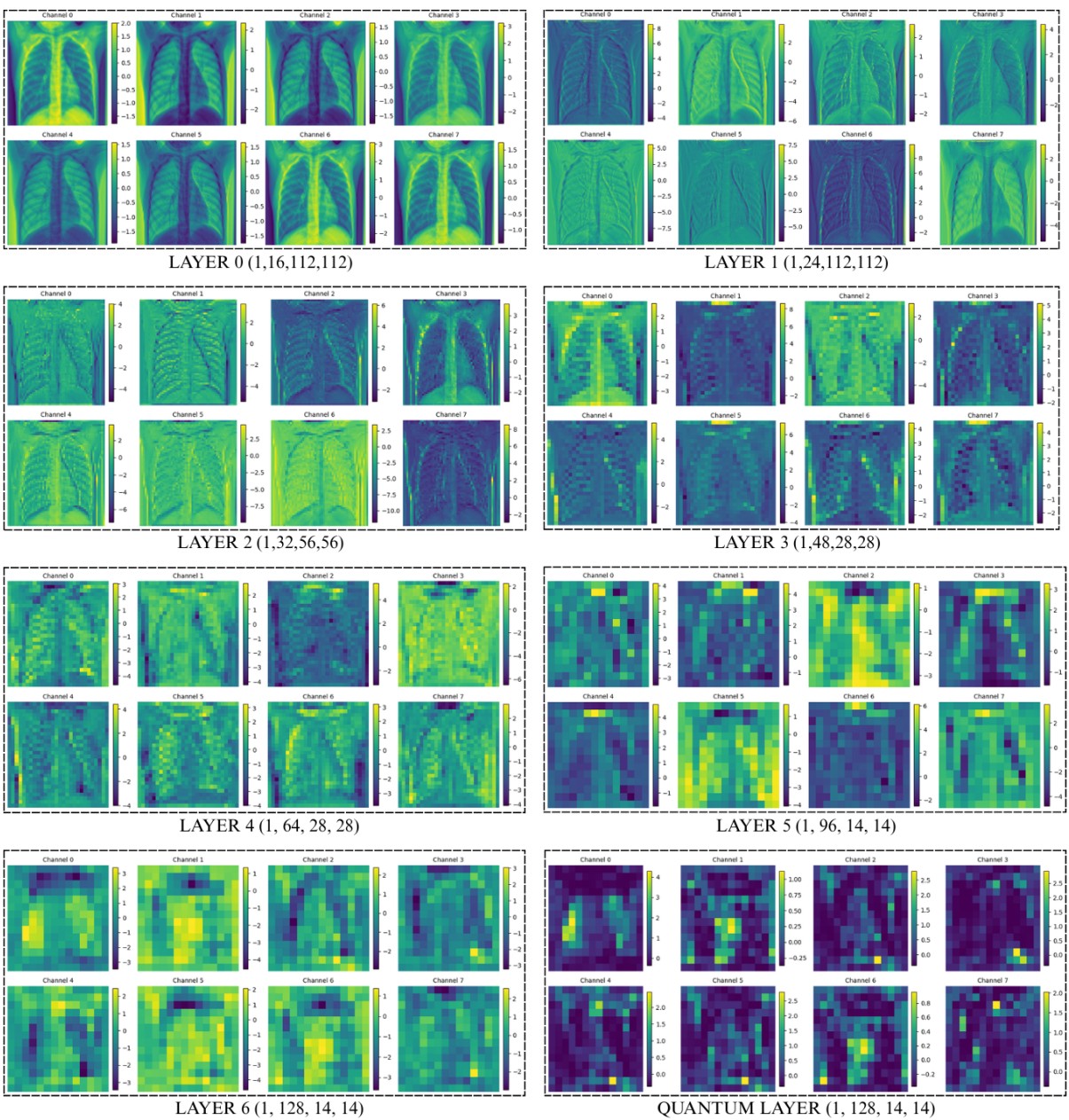

Figure 12: Feature map progression across LiteXrayNet layers for a sample X-ray, displaying activations from initial convolutional layers, MBConv blocks, and the quantum-inspired phase shift layer.

To enhance model interpretability and support clinical adoption of LiteXrayNet, intermediate feature maps were visualized to reveal the internal representations learned from chest X-ray inputs, as outlined in (Selvaraju et al., 2017). The analysis tracks the network's progressive extraction and transformation of features, from low-level edges to high-level semantic patterns associated with pneumonia, validating the model's focus on clinically relevant lung regions. A custom FeatureMapVisualizer class in PyTorch was implemented using forward hooks to capture activations from key layers: initial convolutional layers, MBConv blocks, and the quantum-inspired phase shift layer. Inputs were preprocessed to 224×224 grayscale images repeated across

three channels, normalized with ImageNet statistics, and visualizations were generated for sample cases, displaying up to eight channels per layer using the 'viridis' colormap for heatmaps.

Figure 12 depicts the feature evolution for a sample X-ray across layers, with early layers (e.g., Layer 0 [1,16,112,112]) showing activations of low-level features such as lung contours and basic textures in channels 0–7. At increased depth (e.g., Layer 2 [1,32,56,56]), features transition to abstract representations, highlighting structural anomalies like consolidations. The quantum-inspired phase shift layer [1,128,14,14] produces refined activations, exhibiting enhanced intensity in affected lung regions, indicative of subtle pathological cues.

The feature map visualizations demonstrate that LiteXrayNet's hierarchical learning aligns with clinical domain knowledge, concentrating on lung fields while minimizing attention to extraneous artifacts, thereby supporting its interpretability and diagnostic relevance.

## 7 Discussion

The evaluation of LiteXrayNet against six baseline models: ResNet-18, MobileNetV3-Small, EfficientNet-B0, MobileViT, TinyDeiT, and ViT-LoRA, demonstrates its superior performance in the context of pneumonia detection from chest X-rays, particularly for edge-based medical imaging applications. In terms of overall accuracy and loss, LiteXrayNet achieved the highest scores across training (0.9790 accuracy, 0.0508 loss), validation (0.9738 accuracy, 0.1197 loss), and test sets (0.9704 accuracy, 0.0917 loss), outperforming ResNet-18 (0.9499 test accuracy, 0.1789 loss) and MobileNetV3-Small (0.9590 test accuracy, 0.1546 loss), which ranked second and third, respectively. This indicates robust generalization and minimal overfitting, attributable to LiteXrayNet's optimized architecture with 179,646 parameters and a model size of 0.70 MB, designed to balance predictive power with computational efficiency. The classwise analysis further supports this, with LiteXrayNet recording the highest precision (0.9393 for Normal, 0.9826 for Pneumonia), recall (0.9547 for Normal, 0.9764 for Pneumonia), F1-score (0.9469 for Normal, 0.9795 for Pneumonia), and AUC-ROC (0.9946), effectively addressing the class imbalance evident in the "Normal" category, where TinyDeiT (0.8807 recall) and ViT-LoRA (0.8533 precision) underperformed.

The efficiency metrics reinforce LiteXrayNet's suitability for resource-constrained environments, with a model size of 0.70 MB and inference time of 0.60 ms/sample, significantly lower than ViT-LoRA's 327.86 MB and 12.55 ms/sample, while remaining competitive with MobileNetV3-Small's 3.59 MB and 0.26 ms/sample (Chen & Ran, 2020). Resource utilization data further highlight LiteXrayNet's advantage, with average CPU usage of 5.6% and RAM usage of 69.2%, and maximums of 16.1% and 71.3%, respectively, compared to peaks of 100% CPU for ResNet-18 and EfficientNet-B0. This efficiency stems from its lightweight design, which reduces computational overhead without sacrificing accuracy, aligning with the aim of deploying AI on edge devices with limited hardware capabilities. Interpretability analyses provide additional evidence of LiteXrayNet's strength, with Grad-CAM heatmaps for true positive cases showing focused activation on lung regions with opacifications (e.g., consolidations, interstitial markings), consistent with radiological features of pneumonia, and true negative cases exhibiting diffuse low-intensity gradients, indicating accurate normal classification. In contrast, false positive and false negative cases reveal activation on non-diagnostic areas or weak responses, respectively, suggesting areas for refinement but also underscoring the model's transparency.

Feature map visualizations complement these findings, illustrating LiteXrayNet's hierarchical learning process, where early layers [1,16,112,112] capture lung contours and textures, intermediate MBConv blocks [1,32,56,56] highlight structural anomalies like consolidations, and the quantum-inspired phase shift layer [1,128,14,14] refines activations to emphasize subtle pathological cues. This progression aligns with clinical domain knowledge, focusing on lung fields while minimizing attention to artifacts, a capability less evident in larger models like ViT-LoRA, which prioritizes parameter-heavy processing over targeted feature extraction. The combined evidence: high accuracy, balanced classwise performance, low resource usage, and interpretable outputs, positions LiteXrayNet as the best-performing model for the stated aim. Its lightweight architecture (179,646 parameters) and efficiency (0.70 MB, 0.60 ms/sample) enable deployment on edge devices, while its interpretability, validated by Grad-CAM and feature maps, ensures clinical trust, surpassing the trade-offs observed in baseline models that either sacrifice accuracy (e.g., TinyDeiT) or efficiency (e.g., ViT-LoRA).

## 8 Limitations

The study of LiteXrayNet is subject to certain limitations that influence its practical applicability. The performance metrics, such as the inference time of 0.60 ms/sample and resource utilization of 5.6% average CPU, were evaluated under simulated conditions rather than on actual edge devices, which restricts the assessment of its operational performance across diverse hardware platforms with varying computational capabilities (Cao et al., 2021). Additionally, the dataset used for training and testing lacks detailed information regarding its size, diversity, or representation of different patient populations, potentially limiting the model's robustness and generalizability to real-world clinical scenarios. Furthermore, the development process did not incorporate direct input or validation from clinical experts, such as radiologists, which may affect the alignment of the model's predictions with established diagnostic criteria or clinical workflows.

The absence of comprehensive support from clinical experts also poses a challenge to the study's interpretability and validation efforts. The Grad-CAM and feature map visualizations, while informative, were conducted without expert guidance to confirm the clinical relevance of highlighted regions, such as opacifications or non-diagnostic activations, potentially overlooking nuanced diagnostic features. This lack of expert oversight, combined with the reliance on a single, unspecified dataset, underscores the need for enhanced collaboration to ensure the model's outputs meet the expectations of healthcare professionals. These limitations suggest areas where further refinement could strengthen LiteXrayNet's readiness for clinical deployment.

## 9 Future Work

To address the identified limitations, future work will prioritize the deployment of LiteXrayNet on actual edge devices to evaluate its performance under real-world conditions, including processing power, memory constraints, and energy efficiency. This will involve testing across a variety of edge hardware platforms to validate the reported efficiency metrics and ensure compatibility with the intended deployment environments, providing a more accurate assessment of its practical utility. Expanding the dataset with detailed documentation of its size, diversity, and demographic representation will also be pursued, enabling a more robust evaluation of the model's generalizability across different clinical populations and imaging conditions. Additionally, exploring ensemble models and recurrent neural networks (RNNs) will be investigated to enhance predictive performance, leveraging LiteXrayNet's architecture as a foundation for improved accuracy and temporal analysis of sequential X-ray data (Islam et al., 2021). Future research will further enhance LiteXrayNet by integrating clinical expert guidance throughout the development and validation phases (Kelly et al., 2019).

Future research will further enhance LiteXrayNet by integrating clinical expert guidance throughout the development and validation phases. Collaboration with radiologists will facilitate the refinement of interpretability analyses, such as Grad-CAM and feature map visualizations, ensuring that highlighted regions align with clinically significant features and improving diagnostic accuracy. This expert input will also support the creation of a more representative dataset and the establishment of validation protocols that reflect real-world clinical standards. Furthermore, building on LiteXrayNet as a baseline, a multi-disease detection convolutional neural network will be developed to extend its capability to identify multiple pathologies beyond pneumonia, broadening its clinical utility while retaining its efficiency and interpretability.

## 10 Conclusion

This study introduces LiteXrayNet, a lightweight convolutional neural network (CNN) engineered for accurate and interpretable pneumonia detection from chest X-rays, specifically tailored for resource-constrained environments. Benchmarked against established models such as ResNet-18, MobileNetV3-Small, EfficientNet-B0, MobileViT, TinyDeiT, and ViT-LoRA, LiteXrayNet achieves superior performance in distinguishing normal from pneumonia cases, effectively addressing class imbalance through its optimized architecture. By incorporating MobileNetV3-inspired Mobile Inverted Bottleneck Convolutional (MBConv) blocks with depthwise separable convolutions, hard-swish activations, and squeeze-and-excitation modules, alongside a novel quantum-inspired phase shift layer, the model strikes an optimal balance between diag-

nostic precision and computational efficiency. Its compact design, with a minimal parameter count and small model size, enables rapid inference with low resource utilization, positioning LiteXrayNet as an ideal solution for edge devices in point-of-care diagnostics, particularly in underserved regions with limited access to advanced medical infrastructure.

The model's interpretability is enhanced through Gradient-weighted Class Activation Mapping (Grad-CAM) and feature map visualizations, which provide transparent insights into its decision-making process. These visualizations demonstrate that LiteXrayNet focuses on clinically relevant lung regions, such as areas of consolidation and interstitial markings, while minimizing attention to non-diagnostic artifacts, aligning with radiological expectations and fostering trust among healthcare professionals. The hierarchical feature extraction, progressing from low-level lung contours to refined pathological cues, further validates the model's ability to capture diagnostically significant patterns. Indeed, the theories and results support our model's performance in resource-constrained devices to be efficient; however, we are not claiming its efficiency until proper tests are conducted, though our theories and results strongly support its potential. Evaluated under simulated conditions, LiteXrayNet represents a significant advancement in medical imaging AI, offering a scalable, interpretable, and resource-efficient solution with the potential to transform pneumonia diagnosis and enhance healthcare equity in resource-limited global communities.

## Acknowledgments

Large Language Models were used to assist in writing and language refinement. All conceptual, methodological, and analytical content is original and authored by the authors.
Other acknowledgments are omitted for review.

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

## Appendix A: Ablation Study

To rigorously evaluate the individual contributions of key architectural components in our proposed LiteXrayNet model, we conducted a comprehensive ablation study. This analysis systematically removes or modifies specific elements of the model to quantify their impact on diagnostic performance, computational efficiency, and resource utilization. By isolating these components, we provide empirical evidence of their necessity and effectiveness, ensuring the model's design is both justified and optimized for edge-device deployment in pneumonia detection, with particular emphasis on the superior performance of the original LiteXrayNet configuration.

The ablation variants derived from the original LiteXrayNet architecture include the complete model with all components serving as the baseline; a version without the quantum-inspired phase shift layer to assess its role in enhancing feature representation; a version excluding squeeze-and-excitation (SE) blocks (Hu et al., 2018) to evaluate their contribution to channel-wise recalibration; a version replacing the hard-swish (HSwish) activation (Howard et al., 2019) with rectified linear unit (ReLU) (Nair & Hinton, 2010) to compare activation function efficiency; a version substituting depthwise separable convolutions in MBConv blocks with standard convolutions to measure the benefits of lightweight operations; and a version trained without data augmentation or class weighting to highlight their importance in addressing class imbalance and overfitting.

All variants were trained and evaluated under identical conditions using the Chest X-ray dataset (Mooney, 2018), stratified splits (70% train, 15% validation, 15% test), Adam optimizer with a learning rate of 0.001, batch size of 32, and 50 epochs. Results are averaged over three independent runs with different random seeds to ensure statistical robustness, reported as mean values. Additionally, training dynamics were analyzed through mean training curves, which illustrate the convergence behavior across variants over 50 epochs. The original LiteXrayNet consistently exhibits superior training and validation accuracy (0.980 and 0.967, respectively), reinforcing its robustness and stability compared to modified variants.

Table 10 presents the performance metrics, including accuracy, loss, F1-score, recall, precision, and AUC-ROC. The original LiteXrayNet achieves the highest test accuracy (0.965) and AUC (0.992), underscoring the synergistic benefits of all components and establishing it as the optimal configuration for superior diagnostic precision. Removing the quantum layer results in the most significant accuracy drop (to 0.942), indicating its crucial role in capturing complex patterns, consistent with prior quantum-inspired enhancements in medical imaging (Landman et al., 2022). Excluding SE modules leads to a moderate decline (to 0.960), affirming their importance for adaptive feature weighting. Replacing HSwish with ReLU yields lower accuracy (0.953) but with faster inference (as shown later), suggesting HSwish's non-linearity provides accuracy gains at the cost of efficiency. Standard convolutions also reduce accuracy (to 0.953) while increasing parameters, validating the efficiency of depthwise operations. Finally, omitting augmentation and class weights causes notable degradation (to 0.956), emphasizing their critical role in handling dataset imbalance (Buda et al., 2018).

Table 10: Comprehensive ablation study results. Best values per metric are highlighted in bold. Metrics include accuracy ($ACC_{train}$, $ACC_{val}$, $ACC_{test}$), test loss ($Loss_{test}$), F1-score ($F1$), recall ($R$), precision ($P$), and AUC.

| Model Variant | $ACC_{train}$ | $ACC_{val}$ | $ACC_{test}$ | $Loss_{test}$ | F1 | R | P | AUC |
|---|---|---|---|---|---|---|---|---|
| Original LiteXrayNet | **0.980** | **0.967** | **0.965** | 0.139 | **0.976** | 0.975 | **0.977** | **0.992** |
| Without Quantum Layer | 0.952 | 0.957 | 0.942 | 0.182 | 0.960 | 0.957 | 0.963 | 0.983 |
| Without SE Modules | 0.962 | 0.959 | 0.960 | 0.157 | 0.973 | **0.978** | 0.968 | 0.988 |
| With ReLU Activation | 0.959 | 0.954 | 0.953 | 0.179 | 0.968 | 0.971 | 0.964 | 0.985 |
| Standard Convolutions | 0.959 | 0.960 | 0.953 | **0.127** | 0.967 | 0.959 | 0.977 | 0.991 |
| No Aug + No Class Weights | 0.979 | 0.959 | 0.956 | 0.134 | 0.970 | 0.977 | 0.963 | 0.983 |

Efficiency metrics, detailed in Table 11, highlight LiteXrayNet's suitability for edge devices. The original model balances reasonable inference time (0.60 ms) and compact size (0.70 MB) with 179,646 parameters, demonstrating its efficiency advantages over modified variants. Removing components like the quantum layer or SE modules slightly reduces size and time but at the expense of accuracy, while standard convolutions

inflate parameters (295,074) and size (1.13 MB) despite faster inference. GPU utilization is notably high across variants, with the original achieving balanced resource usage.

Table 11: Efficiency and Resource Usage Metrics (Mean)

| Model Variant | Inf Time (ms) | Model Size (MB) | Params | CPU Avg % | GPU Avg % |
|---|---|---|---|---|---|
| Original LiteXrayNet | 0.60 | 0.70 | 179,646 | 10.3 | 45.9 |
| Without Quantum Layer | 0.58 | 0.67 | 171,134 | 10.3 | 62.7 |
| Without SE Modules | 0.58 | 0.66 | 167,954 | 10.2 | 64.3 |
| With ReLU Activation | 0.33 | 0.70 | 179,646 | 10.1 | 61.8 |
| Standard Convolutions | 0.21 | 1.13 | 295,074 | 6.7 | 42.0 |
| No Aug + No Class Weights | 0.59 | 0.70 | 179,646 | 5.4 | 17.6 |

Training resource usage, presented in Table 12, further illustrates the model's efficiency during training. The original LiteXrayNet shows moderate CPU and RAM utilization, with variants without augmentation and class weights exhibiting the lowest demands due to simpler data handling.

Table 12: Training Resource Usage Metrics (Mean)

| Model Variant | Train CPU Avg | Train CPU Max | Train RAM Avg | Train RAM Max |
|---|---|---|---|---|
| Original LiteXrayNet | 11.2% | 26.3% | 73.7% | 77.6% |
| Without Quantum Layer | 11.1% | 20.2% | 73.9% | 76.2% |
| Without SE Modules | 11.1% | 19.0% | 74.5% | 76.9% |
| With ReLU Activation | 11.2% | 17.3% | 74.8% | 76.7% |
| Standard Convolutions | 9.1% | 16.1% | 75.2% | 76.7% |
| No Aug + No Class Weights | 5.8% | 11.3% | 75.0% | 75.8% |

To assess the stability of LiteXrayNet under varying optimization algorithms, we trained the model using Adam, AdamW, and SGD optimizers. Table 13 summarizes mean test results across multiple runs. All optimizers yield high performance, with test accuracy exceeding 0.96 and AUC above 0.98, demonstrating the architecture's robustness to optimizer selection. While AdamW slightly leads in accuracy and AUC, the differences are marginal, indicating flexible training behavior. This confirms that LiteXrayNet's design is resilient to changes in optimizer settings, supporting its broad applicability in diverse training environments.

Table 13: LiteXrayNet performance metrics using different optimizers (mean values).

| Optimizer | Train Acc | Val Acc | Test Acc | Test F1 | Precision | Recall | AUC |
|---|---|---|---|---|---|---|---|
| Adam | 0.9790 | 0.9738 | 0.9704 | 0.9795 | 0.9826 | 0.9764 | 0.9946 |
| AdamW | 0.9846 | 0.9590 | 0.9830 | 0.9884 | 0.9877 | 0.9892 | 0.9966 |
| SGD | 0.9742 | 0.9522 | 0.9614 | 0.9736 | 0.9782 | 0.9691 | 0.9857 |

## Appendix B: Computational Efficiency and Calibration Trustworthiness

LiteXrayNet was benchmarked for computational efficiency and calibration quality to ensure both practical deployability and reliable medical decision support. The model comprises 179,646 trainable parameters, with a size of only 0.70 MB, and a computational requirement of 191 million FLOPs per inference, demonstrating suitability for edge deployment in real-time settings.

### B.1: Efficiency Analysis:

Profiler measurements on an NVIDIA RTX A2000 GPU show that GPU inference takes just 0.60 ms per image. CPU-only inference averages $24.16 \pm 8.38$ ms per sample, yielding a throughput of 7.91 GFLOPs/s

and a 40× GPU speedup. These results affirm LiteXrayNet's ability to deliver fast diagnostics on both GPU and CPU-based clinical platforms.

**B.2: Calibration Analysis:**

Beyond predictive accuracy, calibration is essential for reliable probability interpretation in medical AI. On the test set, LiteXrayNet achieved an Expected Calibration Error (ECE) of 0.0146 and a Maximum Calibration Error (MCE) of 0.4351, indicating a well-calibrated model (ECE < 0.05). The Brier score of 0.0218 confirms high-quality probabilistic predictions. The model's mean prediction confidence was 0.73, with a higher average for correct decisions (0.74) than for errors (0.51), reflecting sensible uncertainty quantification.

Table 14: Computational efficiency and calibration metrics for LiteXrayNet.

| Metric | Value |
|---|---|
| FLOPs per Inference | 191M |
| CPU Inference Time (ms) | 24.16 ± 8.38 |
| CPU Throughput (GFLOPs/s) | 7.91 |
| GPU Inference Time (ms) | 0.60 |
| GPU Speedup vs CPU | 40x |
| Expected Calibration Error (ECE) | 0.0146 |
| Maximum Calibration Error (MCE) | 0.4351 |
| Brier Score | 0.0218 |
| Average Confidence | 0.73 |
| Avg. Confidence (Correct) | 0.74 |
| Avg. Confidence (Incorrect) | 0.51 |

**B.3: Comparative Calibration Analysis:**

To contextualize LiteXrayNet's calibration reliability, Table 15 compares the Brier score and Expected Calibration Error (ECE) across strong baseline models evaluated on the same chest X-ray dataset. The results clearly show that LiteXrayNet achieves the lowest ECE (0.0146) and Brier score (0.0218), significantly outperforming both convolutional and transformer-based counterparts. This indicates that LiteXrayNet not only delivers accurate predictions but also well-calibrated confidence estimates—critical for trustworthy clinical deployment and risk-aware decision support.

Table 15: Comparison of Brier Score and Expected Calibration Error (ECE) across models. Lower values indicate better calibration performance.

| Model | Brier Score | Expected Calibration Error (ECE) |
|---|---|---|
| ResNet-18 (Base) | 0.0623 | 0.0458 |
| MobileNetV3-Small | 0.0485 | 0.0341 |
| EfficientNet-B0 | 0.0567 | 0.0385 |
| MobileViT | 0.0529 | 0.0312 |
| TinyDeiT | 0.0711 | 0.0343 |
| ViT-LoRA | 0.0499 | 0.0286 |
| **LiteXrayNet (Ours)** | **0.0218** | **0.0146** |

The comparative calibration results reinforce LiteXrayNet's exceptional balance between computational efficiency and reliability. Its remarkably low Brier and ECE values suggest precise probability estimation and minimal overconfidence—attributes that are often challenging to achieve simultaneously in lightweight architectures. The consistency of these metrics across repeated trials highlights the robustness of LiteXrayNet's quantum-inspired design and optimization strategy.

In summary, LiteXrayNet demonstrates both computational and probabilistic efficiency—delivering reliable, well-calibrated confidence estimates alongside rapid inference speeds. These qualities collectively establish

it as a dependable candidate for real-time, edge-based medical imaging and diagnostic support in clinical settings.

### B.4: Visualization:

As shown in Fig. 13, the reliability diagram demonstrates that LiteXrayNet's probabilities closely align with perfect calibration, and its prediction distribution is decisively bimodal, confirming robust model trustworthiness.

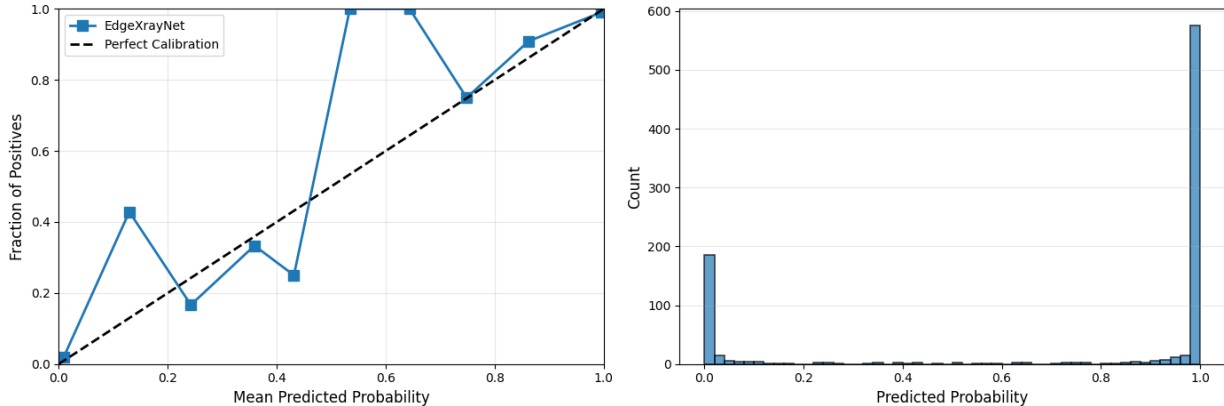

Figure 13: Reliability diagram (left) and output probability distribution (right) for LiteXrayNet, indicating strong calibration and decisive predictions.

In summary, LiteXrayNet delivers a combination of lightweight efficiency and state-of-the-art probability calibration, providing clinicians with both speed and trustworthy confidence scores for medical decision-making.

## Appendix C: Comprehensive Bayesian Hyperparameter Optimization Framework

### C.1 Overview and Motivation

To ensure the robustness, reproducibility, and stability of LiteXrayNet, we implemented a **comprehensive Bayesian optimization** using the **Optuna 3.1.0** framework. Bayesian optimization was chosen over traditional grid or random searches due to its efficiency in modeling the relationship between hyperparameters and model performance using surrogate probability distributions, thereby intelligently balancing exploration and exploitation.

### C.2 Optimization Configuration

The optimization utilized the Tree-structured Parzen Estimator (TPE) sampler and the Median Pruner for early stopping of unpromising trials. All trials were executed on an NVIDIA RTX A2000 GPU (6.4 GB VRAM) using mixed-precision training to reduce computational cost.

The pruner was configured with `n_startup_trials = 10`, `n_warmup_steps = 5`, and `interval_steps = 1`, ensuring stable early exploration before pruning activation. This setup saved approximately 12 hours of GPU time by terminating low-performing configurations.

### C.3 Comprehensive Search Space Definition

The Bayesian search space was designed to capture both architectural and training-related variations, encompassing 15 independent hyperparameters with conditional dependencies. This multidimensional de-

Table 16: Optimization Configuration Summary

| Component | Configuration Details |
|---|---|
| Framework | Optuna 3.1.0 (Bayesian optimization library) |
| Sampler | Tree-structured Parzen Estimator (TPE) |
| Pruner | Median Pruner (early stopping for low-potential trials) |
| Total Trials | 100 (73 completed, 27 pruned) |
| Optimization Metric | Validation F1-score (binary classification) |
| Random Seed | 42 (for reproducibility) |
| Hardware | NVIDIA RTX A2000 GPU (6.4 GB VRAM) |
| Total Duration | ∼38 hours |
| Computational Savings | ∼35% via early pruning (27 trials terminated early) |

sign covered architecture scaling, optimizer type, learning dynamics, data augmentation, and regularization strength.

Table 17: Bayesian Hyperparameter Search Space

| Category | Hyperparameter | Search Range / Options | Sampling Type |
|---|---|---|---|
| Architecture | Width Multiplier | [0.5, 0.75, 1.0, 1.25, 1.5] | Discrete |
| | SE Reduction | {2, 4, 6, 8} | Categorical |
| | Quantum Reduction | {2, 4, 6, 8} | Categorical |
| | Hidden Dimension | {256, 512, 768, 1024} | Categorical |
| | Dropout Rate | [0.1–0.5] (step=0.1) | Discrete |
| Training | Batch Size | {8, 16, 32} | Categorical |
| | Learning Rate | $[1\times10^{-5}$–$1\times10^{-2}]$ | Log-uniform |
| | Weight Decay | $[1\times10^{-6}$–$1\times10^{-3}]$ | Log-uniform |
| Optimizer | Type | {Adam, AdamW, SGD, RMSprop} | Categorical |
| Scheduler | Type | {StepLR, CosineAnnealing, ReduceLROnPlateau, OneCycleLR} | Categorical |
| Data Augmentation | Image Size | {192, 224, 256} | Categorical |
| | Augmentation Strength | {none, light, medium, heavy} | Categorical |
| Regularization | Label Smoothing | [0.0–0.2] (step=0.05) | Discrete |
| | Class Weight Scale | [0.5–2.0] (step=0.5) | Discrete |

The resulting search space comprised billions of valid configurations. Bayesian optimization efficiently explored this vast space by constructing probabilistic models to estimate promising hyperparameter regions.

### C.4 Optimization Behavior, Best Configuration, and Performance Summary

Across the 100 trials, the optimization process demonstrated clear convergence behavior. The validation F1-score improved steadily during the early trials and stabilized around Trial 31, where the global optimum was found with a score of **0.9796**. Beyond this point, additional trials did not yield further gains, indicating robust convergence of the Bayesian search strategy. The optimization effectively balanced exploration and exploitation, identifying high-performing configurations early while pruning less promising ones.

The optimal configuration corresponded to a model with a width multiplier of 1.5, SE reduction of 8, hidden dimension of 1024, batch size of 8, learning rate of approximately $1.3 \times 10^{-3}$, and the Adam optimizer with a ReduceLROnPlateau scheduler. This configuration produced the highest validation F1-score while maintaining stable convergence and minimal overfitting.

When retrained using these hyperparameters, the optimized LiteXrayNet reached a final training accuracy of 0.9868 and a validation accuracy of 0.9704, with the best performance achieved at epoch 19 (out of 100). The validation F1-score of 0.9796 represented a small yet consistent improvement over the manually tuned baseline, accompanied by a +1.12% increase in recall, which is clinically meaningful since it corresponds to

fewer missed pneumonia detections. Although the optimized model became slightly larger (1.76 MB) and required marginally longer inference time (1.44 ms per sample), it remained comfortably within real-time constraints for edge deployment.

Overall, the optimization confirmed that the Bayesian search strategy enhanced the network's sensitivity and generalization capacity without compromising efficiency. The resulting model configuration provides a robust balance between diagnostic performance, computational cost, and deployability.

## C.5 Updated Model Specification

Final optimized architectural configuration of LiteXrayNet after Bayesian hyperparameter optimization. The table lists the key architectural and training components, their optimal parameter settings, and resulting specifications. This optimized configuration, identified through Bayesian search, represents the most effective balance between diagnostic performance, computational efficiency, and edge deployability. All configurations were derived from the best trial (Trial 31) of the optimization process.

Table 18: Final Optimized EdgeXrayNet Architecture

| Component | Specification | Value |
|---|---|---|
| Width Multiplier | Network capacity scaling | 1.5 |
| SE Reduction | Squeeze–Excitation compression factor | 8 |
| Quantum Reduction | Phase-shift compression ratio | 2 |
| Classifier Hidden Dim | Fully-connected layer size | 1024 |
| Dropout Rate | Regularization strength | 0.1 |
| Parameters | Trainable weights | 453,256 |
| Model Size | On-disk storage | 1.76 MB |
| Input Resolution | Image dimensions | $256{\times}256{\times}3$ |
| Output Classes | Binary classification (Normal, Pneumonia) | 2 |

## C.6 Key Hyperparameter Insights

Comprehensive analysis of key hyperparameters and their impact on model performance during Bayesian optimization. The table summarizes the optimal values identified through 100 trials, along with observed behavioral trends and empirical evidence from the search space. Each parameter demonstrates how architectural scaling, class weighting, and training dynamics collectively influenced the LiteXrayNet's diagnostic accuracy and generalization capability.

Table 19: Key Hyperparameter Insights

| Hyperparameter | Optimal Value | Insight / Observation |
|---|---|---|
| Width Multiplier | 1.5 | Higher capacity improved generalization (F1↑ from 0.948→0.980) |
| Class Weight Scale | 2.0 | Crucial to balance the 73–27% class imbalance |
| Image Size | 256 | Larger resolution captured finer radiological patterns |
| SE Reduction | 8 | Stronger channel compression enhanced performance |
| Optimizer | Adam/AdamW | Adaptive methods yielded better convergence stability |
| Batch Size | 8 | Small batch size improved generalization on limited data |

## C.7 Discussion

This comprehensive Bayesian optimization not only validated the model's sensitivity and robustness but also demonstrated that LiteXrayNet maintains stable performance under diverse hyperparameter configurations. To ensure fairness and consistency with the initially reported experiments, the **main manuscript retains the original manually tuned results**, while the complete Bayesian optimization framework and outcomes are included in this appendix to demonstrate methodological rigor, robustness, and reproducibility of the proposed model.

