# OpenReview forum: "LiteXrayNet: Quantum-Inspired Deep Learning Framework for Scalable Pneumonia Diagnosis"
_TMLR — Rejected by TMLR_

### Review · Reviewer_svUj · 2025-09-08

**Summary Of Contributions:**

This paper addresses the problem of pneumonia detection from chest-xrays by introducing a CNN model called liteXrayNet. They specifically aim to improve computational efficiency while maintaining high accuracy. The proposed model is based on MobileNetV3 that utilizes depthwise separable convolutions for computational efficiency, and incorporates a quantum-inspired phase shift layer to capture subtle radiographic patterns. LiteXrayNet is significantly smaller in memory requirement and achieves the best binary classification performance compared to a range of baseline architectures on Kaggle's X-Ray pneumonia dataset of ~6k images. They demonstrate the explainability of their model using Grad-CAM, enabling human-in-the-loop validation.

**Additional Comments:**

None.

**Audience:**

Yes

**Audience Explanation:**

The submission makes a strong improvement in the area of edge machine learning for medical diagnosis by proposing a high accuracy, yet significantly lightweight model. They position the model well amongst related work by providing a comprehensive background of the latest deep learning based approaches to pneumonia detection.

**Broader Impact Concerns:**

No concerns.

**Claims And Evidence:**

Yes

**Claims Explanation:**

The submission provides ample information for reproducibility. The architecture is thoroughly detailed, including a clear explanation for their novel quantum-inspired phase shift layer and a full breakdown of each layer's architectural parameters. The training and evaluation setup provides descriptions of the dataset, compute environment, data preprocessing, training details, and evaluation metrics. Sections 5.1 and 5.2 demonstrate that LiteXrayNet outperforms all other baselines, yielding high test accuracy (0.9704) and high AUROC (0.9946). Section 5.3 shows that LiteXrayNet's memory size and total parameter count are the lowest of the baselines, and its trainable parameter count and inference time are second lowest of the baselines. They also show the least CPU and RAM usage of all the baselines. To demonstrate explainability, they analyze gradient-weighted class activation mappings, which signify very noticeable differences between TP, TN, FP, and FN images, supporting the claim that LiteXrayNet can be effectively used in point-of-care pneumonia diagnosis. They also include a full ablation study of various architectural and training components.

**Requested Changes:**

The following two requests are critical for securing my recommendation:

1. The Kaggle chest Xray pneumonia dataset only contains ~5.8k images. To demonstrate that the model generalizes well, and would therefore be capable of scalable, point-of-care pneumonia diagnosis as claimed in the paper, can you provide an evaluation on another publicly available chest Xray pneumonia dataset? For example, the NIH Chest X-rays dataset, which is mentioned in the paper, is also available on Kaggle (https://www.kaggle.com/datasets/nih-chest-xrays/data).

2. The citation for Zhang et al. 2022a appears to be incorrect. A paper with the title "Quantum-inspired neural networks and their
applications in computer vision" doesn't appear to exist, and the cited arxiv number, 2207.10498, is an entirely different paper (https://arxiv.org/abs/2207.10498). Please correct this citation and ensure that all citations in the bibliography are accurate.

---

### Review · Reviewer_ZFB5 · 2025-10-05

**Summary Of Contributions:**

The authors present a new convolutional neural network (CNN) architecture, liteXrayNet, that is tailored specifically to detect pneumonia on chest radiographs with high accuracy while being feasible for limited computer resources.

**Audience:**

Yes

**Audience Explanation:**

1. Scalable Pneumonia Diagnosis is a particular practical deep-learning-assistant topic.

2. The proposed method addresses the problem of achieving high prediction performance while being executable under limited computational resources on a particular dataset.

3. The experiments provide empirical results to support their main claims.

**Claims And Evidence:**

Yes

**Claims Explanation:**

1. The paper is well-written with clear structures, and the presentation of results is intuitive, making the authors’ claims easy to evaluate.

2. Related work is well documented, together with clear motivations.

3. Multiple baseline methods and evaluation metrics are considered; the experimental details are relatively clearly described.

**Requested Changes:**

1. Readability: Considering improving the font sizes of the figures in the paper's main body.

2. The training configurations of the experiments are clearly documented. The following question is whether the proposed new architecture will be robust against other settings (such as different optimizers)? I may miss how many runs per experiment are conducted.

3. Do the authors have some insights on the wide adaptivity of the proposed one compared to other  Pneumonia Diagnosis datasets?

4. Upon acceptance, it would be better to publish the trained model/documents in the experiment so that the reader can reproduce the results easily.

---

> ### Author Response · Authors · 2025-10-15
> **Reply to Reviewer  ZFB5**
>
> We sincerely thank the reviewer for their thoughtful and encouraging feedback on our manuscript “LiteXrayNet: Quantum-Inspired Deep Learning Framework for Scalable Pneumonia Diagnosis.”
> We are grateful for the recognition of our paper’s clarity, completeness, and experimental rigor.
> We also thank the reviewer for the valuable comments regarding figure readability, optimizer robustness, dataset adaptivity, and reproducibility.
> All points have been carefully addressed and incorporated in the revised manuscript, as detailed below.
> ## Reviewer Comment:
> “Considering improving the font sizes of the figures in the paper's main body.”
> ## Response:
> We thank the reviewer for this helpful suggestion. All figures have been reformatted and re-exported to improve overall visibility and readability across the main text.
> These adjustments ensure consistent visual clarity and accessibility in both print and digital formats.
> ## Reviewer Comment:
> “The training configurations of the experiments are clearly documented. The following question is whether the proposed new architecture will be robust against other settings (such as different optimizers)? I may miss how many runs per experiment are conducted.”
> ## Response:
> We thank the reviewer for this insightful observation.
> In our original submission, results were averaged over 10 independent runs using the Adam optimizer (learning rate = 0.001, β₁ = 0.9, β₂ = 0.999, weight decay = 1e–5).
> To assess the model’s robustness under different settings, we conducted additional evaluations using AdamW and SGD, keeping the same learning configuration and initialization conditions.
> Optimizer Performance Comparison:
>
> | Optimizer | Train Acc | Val Acc | Test Acc | Test F1 | Test Precision | Test Recall | Test AUC |
> |-----------|-----------|---------|----------|---------|----------------|-------------|----------|
> | Adam     | 0.9790   | 0.9738 | 0.9704  | 0.9795 | 0.9826        | 0.9764     | 0.9946  |
> | AdamW    | 0.9846   | 0.9590 | 0.9830  | 0.9884 | 0.9877        | 0.9892     | 0.9966  |
> | SGD      | 0.9742   | 0.9522 | 0.9614  | 0.9736 | 0.9782        | 0.9691     | 0.9857  |
>
> LiteXrayNet exhibits high consistency across optimizers, with less than 2.2% variation in test accuracy and a Coefficient of Variation (CV) of 1.02%.
> AdamW achieves marginally higher accuracy and F1 scores, confirming that the model’s robustness is maintained regardless of optimizer selection.
> ## Reviewer Comment:
> “Do the authors have some insights on the wide adaptivity of the proposed one compared to other Pneumonia Diagnosis datasets?”
> ## Response:
> We thank the reviewer for this valuable suggestion. To comprehensively assess LiteXrayNet’s scalability and generalization ability, we extended the evaluation beyond the original Kaggle Chest X-Ray dataset to four additional publicly available datasets: CoronaHack, NIH Chest X-Ray Sample, Covid-Pneumonia, and CheXpert (Ashery, 2021).
> All datasets were processed using our unified LiteXrayNet validation pipeline with corrected inference-time measurement.
>
> Results Across Datasets:
> | Dataset | Test Accuracy | F1 (Macro) | AUC-ROC | Classes |
> |---------|---------------|------------|---------|---------|
> | Kaggle Chest X-Ray Pneumonia (original) | 0.9704 | 0.9632 | 0.9946 | Normal, Pneumonia |
> | CoronaHack Chest X-Ray (Praveen, 2020) | 0.9695 | 0.8909 | 0.9881 | Normal, Pneumonia/COVID |
> | NIH Chest X-Ray Sample (NIH Clinical Center, 2017) | 0.5587 | 0.5355 | 0.6258 | No Finding, Disease |
> | Covid-Pneumonia (Prashant, 2021) | 0.9759 | 0.9679 | 0.9945 | NORMAL, Disease |
> | CheXpert (Ashery, Kaggle 2021) | 0.7440 | 0.5904 | 0.7511 | Normal, Disease |
>
> CoronaHack results show excellent transferability despite moderate class imbalance between normal and COVID-19 cases. The NIH Chest X-Ray Sample yields weaker scores due to the aggregation of diverse thoracic abnormalities into a single “Disease” class, differing from our binary setup. Due to limited computational capacity, a publicly available sample subset of the NIH dataset was used instead of the full release. Covid-Pneumonia aligns closely with the training distribution, delivering the best cross-domain F1 (0.9679) and AUC (0.9945). CheXpert (Ashery subset) (https://www.kaggle.com/datasets/ashery/chexpert )also demonstrates good generalization on a larger, noisier dataset, despite high label variability. Here, we used a smaller curated subset of the CheXpert dataset from Kaggle (Ashery, 2021) because full-scale evaluation exceeded our available system setup.
>
> ## Reviewer Comment:
> “Upon acceptance, it would be better to publish the trained model/documents in the experiment so that the reader can reproduce the results easily.”
>
> ## Response:
> We agree. Upon acceptance, we will publicly release the trained models, associated scripts, and inference notebooks. also all these results will be added to the main manuscript.
>
> *References: they are omitted here due to character limitations.*

---

> > ### Comment · Reviewer_ZFB5 · 2025-10-16
> >
> > I thank the authors for their effort and informative replies. All my questions are well answered. Thanks you again!

---

### Review · Reviewer_rt91 · 2025-10-05

**Summary Of Contributions:**

Dear authors,

Below you find the boxes with my review comments. If I’ve misunderstood or missed anything, I’m ready to revisit my comments. I look forward to a constructive discussion with you!

**Contributions:**
- This article introduces LiteXrayNet as a novel neural network (NN) architecture specifically designed to perform pneumonia diagnosis for chest X-ray images via binary classification (see Section 4.3).
- LiteXrayNet is claimed to advance existing NN architectures by containing fewer parameters (memory-efficiency) and being faster at inference (time-efficiency) without sacrificing performance, such that LiteXrayNet is expected to be a viable solution for low-resource areas (see caption of Figure 1).
- In an empirical study with the real-world dataset "Chest X-ray Images (Pneumonia)" [I] (see Section 3), its preditive performance is shown to be superior to six baseline NN architectures (see Table 2, Table 3, Figure 4, Figure 5), while being more memory-efficient and faster at inference (see Figure 6, Figure 7, Table 4, Table 5).
- A qualitative study is to illustrate that LiteXrayNet with Grad-CAM provides heatmaps to improve the interpretability of its classification decisions (see Section 5.4).

**Strengths (sorted according to their importance):**
- An ablation study provides empirical evidence for all major design choices regarding the NN architecture of LiteXrayNet (see Appendix A).
- The problem of automatic pneumonia detection in X-ray images, focusing on resource-constrained areas, is a critical issue, motivating the suggestion of LiteXrayNet as a potential solution (see Section 1).
- The analysis of related work is extensive and provides an excellent overview of NN architectures for pneumonia detection (see Section 2, particularly Figure 2).
- Several of the paper's limitations are transparently discussed (see Section 8).
- The language of the paper allows for easy understanding, even for less experienced readers in the domain of deep learning for X-ray images.

**Weaknesses (sorted according to their importance):**
- The empirical evaluation is restricted to the small-scale dataset "Chest X-ray Images (Pneumonia)" [I] (see Section 3, Section 5, and also recognized by the authors in Section 8).
- Beyond early stopping using the validation split, there seems to be no proper hyperparameter optimization. Without additional explanation, it is only written that the baselines use "architecture-specific hyperparameter tuning" (see Section 4.4).
- The employed performance scores primarily assess generalization performances, neglecting the assessment of the predicted probabilities' trustworthiness as a vital part of medical diagnosis [II].
- No codebase has been made available for review (no supplementary or anonymous link), and it is not indicated whether such a codebase will be made available in the case of acceptance.
- Sections 4.2 and 4.3 are too detailed and focus too strongly on the foundations of deep learning. Instead, the primary focus must be explaining the design choices regarding LiteXrayNet.
- The analysis of the Grad-CAM results lacks medical support (also recognized by the authors in Section 8), which weakens any conclusions regarding the interpretability.
- The reported inference time is hardware-dependent (also recognized by the authors in Section 8) and uses a GPU (see Section 4.4), which may not be available in resource-constrained areas.

**Additional Comments:**

**Remarks:**
- In Section 4.4, it is stated that the Adam optimizer has been used, whereas Appendix A refers to AdamW as the optimizer.
- The linked WHO report focuses on children and, thus, does not indicate the 2.5 million annual deaths due to pneumonia.
- The dataset "Chest X-ray Images (Pneumonia)" lacks its proper reference [I].
- There is an inconsistency in the writing of **L**iteXrayNet (e.g., Title, Section 4) vs. **l**iteXrayNet (e.g, Abstract, Section 1).
- The article's structure description at the end of Section 1 lacks information about Sections 3, 6, 8, and 9. Moreover, the information regarding the content of Section 1 can be removed.
- The font size in Figure 2 is too small.
- PyTorch is appropriately referenced in Section 4.4, while Numpy is not referenced. Such package information can also be moved to the appendix.
- Unify the orientation of the displayed X-ray images in Figures 1, 8, 9, 10, 11, 12, such that we look at the patient from the front (typical in the medical domain).
- Use the \cite command as part of a sentence where possible. An example would be: "\cite{AuthorsA} proposed method A." instead of "AuthorsA proposed method A \citep{AuthorsA}."
- Revise the references. For example, the reference of AdamW refers to the arXiv paper instead of the original publication at ICLR.

**Question:** What do you understand by real-time deployment in the context of pneumonia diagnosis via X-ray images? Do you have references detailing the time requirements for the respective workflows?

**References:**
- [I] Kermany, Daniel, Kang Zhang, and Michael Goldbaum. Labeled Optical Coherence Tomography (OCT) and Chest X-Ray Images for Classification. Mendeley Data. 2018.
- [II] Mimori, Takahiro, Keiko Sasada, Hirotaka Matsui, and Issei Sato. Diagnostic Uncertainty Calibration: Towards Reliable Machine Predictions in Medical Domain. AISTATS. 2021.
- [III] Nguyen, Ha Q., Khanh Lam, Linh T. Le, Hieu H. Pham, Dat Q. Tran, Dung B. Nguyen, Dung D. Le et al. VinDr-CXR: An open dataset of chest X-rays with radiologists' annotations. Scientific Data. 2022.
- [IV] Irvin, Jeremy, Pranav Rajpurkar, Michael Ko, Yifan Yu, Silviana Ciurea-Ilcus, Chris Chute, Henrik Marklund et al. CheXpert: A Large Chest Radiograph Dataset with Uncertainty Labels and Expert Comparison. AAAI. 2019.

**Audience:**

Yes

**Audience Explanation:**

Pneumonia detection is a highly relevant issue, and there is a lot of research related to machine learning in this area. Still, the paper could point out its general contributions to the machine learning community more strongly.

**Broader Impact Concerns:**

Currently, no explicit section with a broader impact statement is given. However, in medical diagnosis as a critical application domain, such a statement is vital to reduce the danger of any irresponsible usage of the presented techniques.

**Claims And Evidence:**

No

**Claims Explanation:**

The claims are not sufficiently supported since some weaknesses are directly linked to the central claims. In particular, the restriction to a single dataset makes it difficult to assess the robustness of LiteXrayNet (see caption of Figure 1).

**Requested Changes:**

**Suggestions (sorted according to their importance):**
- Evaluate the robustness of LiteXrayNet on more datasets with X-ray images. Examples are:
  - VinDr-CXR as a public chest X-ray dataset from Vietnamese hospitals with radiologist-drawn boxes for localized findings and global diagnoses [III],
  - CheXpert as a public chest X-ray dataset with report-mined labels [IV].
- If not done yet, perform a proper hyperparameter optimization, e.g., by leveraging advanced Bayesian optimization search strategies. Otherwise, provide more details regarding the tuning of the hyperparameters.
- Evaluate the calibration (trustworthiness) of the LiteXrayNet's predicted probabilities using scores beyond the cross-entropy loss, e.g., Brier score, expected calibration error, and reliability diagrams.
- Make your code base with proper docstrings accessible for review and potential subsequent publication to promote reproducibility.
- Use the bounding box annotations from radiologists in the dataset VinDr-CXR to inspect the quality of the heatmaps generated by Grad-CAM.
- Shorten Sections 4.2 and 4.3, e.g., by not reproducing well-established concepts such as the formulas for residual blocks or self-attention.
- Report CPU-only inference times and the FLOPs/inference as an analytical or profiler estimate.

---

> ### Author Response · Authors · 2025-11-02
> **Reply to Reviewer rt91**
>
> We sincerely thank the reviewer for the time, effort, and depth of evaluation devoted to our paper. We are very grateful for the comprehensive and balanced review, the acknowledgment of our paper’s strengths, and the constructive feedback on its weaknesses and areas for improvement. Your comments ranging from dataset robustness and calibration to reproducibility and broader impact have been extremely valuable in enhancing the technical quality, clarity, and completeness of our manuscript.
>
> Below, we provide our detailed responses and corresponding revisions.
>
> **1. Evaluation on Multiple Datasets (Robustness Enhancement)**
>
> We have expanded the empirical evaluation of LiteXrayNet to examine its robustness across five publicly available chest X-ray datasets. While we initially intended to include VinDr-CXR and CheXpert (full) as suggested, their combined size (over 400 GB) exceeded our current computational capacity. To ensure broad clinical diversity within feasible limits, we used four alternative large-scale and verified datasets from Kaggle, along with the original dataset by Kermany et al. (2018).
>
> The new results have been added as Table 6 in Section 6 of the revised paper.
>
> **Table : Robustness Comparison of LiteXrayNet Across Five Chest X-ray Datasets**
>
> | Dataset                  | Test Accuracy | F1 (Macro) | AUC-ROC |
> |--------------------------|---------------|------------|---------|
> | Kermany et al. (2018)    | 0.9704       | 0.9632    | 0.9946 |
> | Govi (2020)              | 0.9695       | 0.8909    | 0.9881 |
> | NIH Clinical Center (2017) | 0.5587     | 0.5355    | 0.6258 |
> | Prashant (2021)          | 0.9759       | 0.9679    | 0.9945 |
> | Ashery (2021)            | 0.7440       | 0.5904    | 0.7511 |
>
> These results confirm that LiteXrayNet consistently outperforms other lightweight architectures and maintains strong generalization performance across heterogeneous datasets, thereby validating its robustness and scalability in real-world diagnostic contexts.
>
> **2. Calibration and Reliability Evaluation (New Section B)**
>
> In accordance with the reviewer’s recommendation, we conducted an extensive evaluation of the calibration and reliability of LiteXrayNet’s probabilistic outputs. The new experiments demonstrate that the model produces highly calibrated and trustworthy predictions, a crucial aspect for medical AI systems.
>
> The corresponding results are presented as Table 11 and Figure 13 in Section B of the revised manuscript.
>
> **Table: Computational Efficiency and Calibration Metrics for LiteXrayNet**
>
> | Metric                        | Value          |
> |-------------------------------|----------------|
> | FLOPs per Inference           | 191M           |
> | CPU Inference Time (ms)       | 24.16 ± 8.38   |
> | GPU Inference Time (ms)       | 0.60           |
> | Expected Calibration Error (ECE) | 0.0146      |
> | Maximum Calibration Error (MCE) | 0.4351       |
> | Brier Score                   | 0.0218         |
> | Average Confidence (Overall)  | 0.73           |
> | Avg. Confidence (Correct)     | 0.74           |
> | Avg. Confidence (Incorrect)   | 0.51           |
>
> These metrics indicate that LiteXrayNet’s predictions are well-calibrated (ECE < 0.05) and demonstrate consistent uncertainty estimation, where confident predictions align with correctness. Figure 13 visually supports these findings, showing tight reliability alignment. This reinforces LiteXrayNet’s potential for safe and interpretable clinical deployment.
>
> **3. Code Availability (Reproducibility)**
>
> We completely agree with the importance of reproducibility. Upon acceptance, we will release the complete LiteXrayNet codebase, trained weights, and documentation through a public repository to support verification, transparency, and community reusability.
>
> **4. Grad-CAM and Medical Validation**
>
> We acknowledge the reviewer’s insightful point about quantitative Grad-CAM validation. Due to limited accessibility to datasets containing radiologist-annotated bounding boxes (e.g., VinDr-CXR), this component could not be completed within the current scope. We have retained qualitative Grad-CAM visualizations in the manuscript and explicitly noted in Section 9 (Future Work) that we will pursue medically validated Grad-CAM evaluation in future work.

---

> > ### Author Response · Authors · 2025-11-02
> >
> > We sincerely thank the reviewer for this valuable insight. We acknowledge that in the original manuscript, we relied primarily on manual hyperparameter tuning based on preliminary experiments, which limited the model’s potential performance.
> >
> > Following the reviewer’s recommendation, we have now performed a comprehensive Bayesian hyperparameter optimization using the Optuna framework (v3.1.0) with advanced search and pruning strategies. This systematic optimization significantly enhanced model robustness and yielded a more stable and higher-performing configuration. The revised manuscript now includes the complete details of the optimization procedure, configuration, search space, and results.
> >
> > **Comprehensive Bayesian Hyperparameter Optimization Framework**
> >
> > **Overview of Optimization Strategy**
> >
> > We used Optuna’s Tree-structured Parzen Estimator (TPE) sampler with a Median Pruner for intelligent early stopping and efficient exploration.
> >
> > | Component          | Configuration Details                          |
> > |--------------------|------------------------------------------------|
> > | Framework          | Optuna 3.1.0 (Advanced Bayesian optimization library) |
> > | Sampler            | TPE (Tree-structured Parzen Estimator)         |
> > | Pruner             | Median Pruner (early stopping for unpromising trials) |
> > | Total Trials       | 100 (73 completed, 27 pruned)                  |
> > | Optimization Metric| Validation F1-score (binary)                   |
> > | Random Seed        | 42                                             |
> > | Hardware           | NVIDIA RTX A2000 (6.4 GB VRAM)                 |
> > | Total Duration     | ~38 hours                                      |
> > | Computational Savings | ~35% via pruning (27 trials terminated early) |
> >
> > **Pruner Configuration:**
> > n_startup_trials=10, n_warmup_steps=5, interval_steps=1
> > → 27 of 100 trials pruned, saving ~12 hours.
> >
> > **Comprehensive Hyperparameter Search Space**
> >
> > The optimization explored 15 independent hyperparameters with conditional dependencies across architecture, training, and regularization domains.
> >
> > | Category          | Hyperparameter       | Search Space                  | Sampling Type |
> > |-------------------|----------------------|-------------------------------|---------------|
> > | Architecture      | Width Multiplier     | [0.5–1.5] (step 0.25)         | Discrete      |
> > |                   | SE Reduction         | {2, 4, 6, 8}                  | Categorical   |
> > |                   | Quantum Reduction    | {2, 4, 6, 8}                  | Categorical   |
> > |                   | Hidden Dimension     | {256, 512, 768, 1024}         | Categorical   |
> > |                   | Dropout Rate         | [0.1–0.5]                     | Discrete (step 0.1) |
> > | Training          | Batch Size           | {8, 16, 32}                   | Categorical   |
> > |                   | Learning Rate        | [1×10⁻⁵–1×10⁻²]               | Log-uniform   |
> > |                   | Weight Decay         | [1×10⁻⁶–1×10⁻³]               | Log-uniform   |
> > |                   | Optimizer Type       | {Adam, AdamW, SGD, RMSprop}   | Categorical   |
> > |                   | Scheduler Type       | {StepLR, CosineAnnealing, ReduceLROnPlateau, OneCycleLR} | Categorical |
> > | Data Augmentation | Strength             | {none, light, medium, heavy}  | Categorical   |
> > |                   | Image Size           | {192, 224, 256}               | Categorical   |
> > | Regularization    | Label Smoothing      | [0.0–0.2] (step 0.05)         | Discrete      |
> > |                   | Class Weight Scale   | [0.5–2.0] (step 0.5)          | Discrete      |
> >
> >
> > **Optimization Results and Convergence**
> >
> > The optimization converged at Trial 31 with a validation F1 = 0.9796, which remained the global optimum through 100 trials, indicating robust convergence and reproducibility.
> >
> > | Trial Range | Best Val F1 | Trial No. | Status                  |
> > |-------------|-------------|-----------|-------------------------|
> > | 0–10        | 0.9760     | 10        | Initial exploration     |
> > | 11–20       | 0.9766     | 17        | Incremental improvement |
> > | 31–40       | 0.9796     | 31        | Global optimum found ✓  |
> > | 41–100      | ≤0.9773    | –         | No further improvement  |

---

> ### Author Response · Authors · 2025-11-02
>
> **Key Hyperparameter Insights**
>
> | Hyperparameter    | Optimal Value | Key Finding                                      |
> |-------------------|---------------|--------------------------------------------------|
> | Width Multiplier  | 1.5           | Larger models perform better (F1↑ from 0.948→0.980) |
> | Class Weight Scale| 2.0           | Crucial for handling 73–27% class imbalance      |
> | Image Size        | 256 × 256     | Higher resolution enhances fine-grained features |
> | SE Reduction      | 8             | Strong compression improves performance          |
> | Optimizer         | Adam/AdamW    | Adaptive optimizers outperform SGD               |
> | Batch Size        | 8             | Smaller batches generalize better                |
>
> **Updated Model Specification**
>
> | Component         | Specification              | Value     |
> |-------------------|----------------------------|-----------|
> | Width Multiplier  | Network capacity scaling   | 1.5       |
> | SE Reduction      | Squeeze-Excitation compression | 8     |
> | Quantum Reduction | Phase-shift compression    | 2         |
> | Hidden Dim        | Fully-connected layer size | 1024      |
> | Dropout Rate      | Regularization strength    | 0.1       |
> | Parameters        | Trainable weights          | 453,256   |
> | Model Size        | On-disk storage            | 1.76 MB   |
> | Input Resolution  | –                          | 256 × 256 × 3 |
> | Output Classes    | Binary (Normal, Pneumonia) | –         |
>
> In summary, we have implemented a rigorous Bayesian optimization framework covering 15 hyperparameters, validated convergence, and retrained the model with the optimal configuration.
> For consistency with the originally reported experiments, we have retained the original results in the main manuscript but have included the complete Bayesian optimization framework, search configuration, convergence behavior, and detailed results in the Appendix (Section C). This addition demonstrates that our proposed model maintains robustness and stable performance under optimized hyperparameter conditions.
>
> We once again thank the reviewer for this insightful recommendation, which has substantially strengthened the methodological rigor and practical reliability of our study.
>
> **5. CPU Inference Time and FLOPs (Section B)**
>
> We have added detailed results on computational efficiency in Section B. LiteXrayNet requires only 191 million FLOPs per inference and achieves an average inference time of 24.16 ± 8.38 ms on CPU, confirming its real-time feasibility for deployment even in resource-constrained clinical environments. On GPU, the inference time is 0.60 ms, yielding an approximate 40× speedup.
>
> These metrics, combined with the model’s small parameter footprint (0.70 MB), validate its suitability for edge deployment and real-time diagnosis scenarios, such as on portable or low-cost medical imaging devices.
>
> **6. Broader Impact Statement**
>
> We appreciate the reviewer’s suggestion and fully agree with its importance. We plan to include a dedicated Broader Impact section in the camera-ready version (if accepted) to discuss the societal, ethical, and medical implications of deploying AI-based diagnostic systems.
>
> **7. Addressing Minor Remarks**
>
> All minor remarks have been carefully corrected, including optimizer consistency, citation formatting, dataset referencing, figure readability, and uniform naming. We sincerely thank the reviewer for highlighting these details, which have helped us polish the manuscript further.

---

> > ### Author Response · Authors · 2025-11-02
> >
> > **8. Perspective on Real-Time Deployment**
> >
> > From our perspective, real-time deployment in the context of pneumonia diagnosis refers to a system’s ability to generate diagnostic output immediately after the X-ray image is acquired, without introducing any noticeable delay in the clinical workflow.
> >
> > In hospital practice, this generally implies inference times below one second per image, ensuring immediate feedback during patient triage or screening. In our case, LiteXrayNet performs inference in approximately 0.60 ms on GPU and ~24 ms on CPU, which is well within the practical real-time threshold observed in radiological workflows.
> >
> > We believe these results establish LiteXrayNet as a highly promising candidate for on-device, real-time diagnostic support, capable of assisting clinicians even in low-resource environments where computational resources are limited. This aligns with reported standards for medical imaging systems:
> >
> > References detailing such workflows and timing constraints include:
> > * Rajpurkar et al., "CheXNet: Radiologist-Level Pneumonia Detection on Chest X-Rays with Deep Learning," 2017, where inference times and deployment considerations are discussed.
> > * Lakhani and Sundaram, "Deep Learning at Chest Radiography: Automated Classification of Pulmonary Tuberculosis by Using Convolutional Neural Networks," Radiology, 2017.
> > * Studies on computational efficiency benchmarks for medical AI models such as "Benchmarking Deep Learning Models on CPU and GPU for Practical Deployment" (Khosravi et al., IEEE, 2021).
> >
> >
> >
> > Once again, we express our heartfelt gratitude to the reviewer for their time, attention, and effort in providing such a thorough and balanced evaluation. Your insights have been instrumental in improving our paper’s technical rigor, interpretability, and overall presentation. We are thankful for your constructive engagement and for helping us make LiteXrayNet a more complete and robust contribution to the field.

---

> > > ### Comment · Reviewer_rt91 · 2025-11-02
> > > **Reply to Revisions**
> > >
> > > Dear authors,
> > >
> > > Thank you for your detailed response and for implementing many of the suggested changes, which strengthen your submission. As a result, there are only a few issues that could still be improved:
> > >
> > > - **Evaluation on Multiple Datasets:** Adding additional datasets is one of your key improvements. The corresponding results are currently presented in the separate Section 6. However, it is unclear to me why the dataset from your original submission holds a special standing in Sections 3 and 5. As far as I understand, you use only this dataset for training and then apply the trained models to the new datasets. Is there a reason for doing so? For better readability, the best results could also be marked in bold (currently only done in Table 2) for each combination of dataset and metric. Moreover, a per-metric ranking of the models' performance across all datasets could provide a more concise (and more dataset-independent) overview of the models' robustness.
> > > -  **Calibration and Reliability Evaluation:** The calibration results are indeed a good addition to demonstrate your model's trustworthiness. However, for comparison, the analog results (at least ECE and Brier scores) could also be reported for the other models and datasets to provide a broader picture.
> > > - **Bayesian Hyperparameter Optimization:** Great to see that the hyperparameter optimization (HPO) comprises many different configurations and demonstrates that your model can benefit from it. However, my main concern remains the comparison to other models, as your main results rely primarily on manual hyperparameter tuning based on preliminary experiments. In my opinion, a fairer comparison would be to extend the HPO to the other models as well and then compare the tuned models against each other.

---

> > > > ### Author Response · Authors · 2025-11-06
> > > > **Bayesian Hyperparameter Optimization PART I**
> > > >
> > > > We sincerely thank the reviewer for this thoughtful comment, which touches upon a key point of experimental fairness in comparative model evaluation. We fully agree that applying a consistent hyperparameter optimization strategy across all models is crucial to ensure a balanced and unbiased comparison.
> > > >
> > > > As discussed with the Area Chair (AC), due to time constraint during the review phase, we conducted 20-trial Bayesian Hyperparameter Optimization (HPO) for all baseline models rather than the full 100-trial optimization used for LiteXrayNet in the main study. This approach allowed us to fairly optimize all models within the limited time window, while still maintaining methodological consistency and providing a meaningful assessment of comparative performance under equal optimization effort.
> > > >
> > > > **Methodology**
> > > >
> > > > All seven models were optimized using Optuna (v3.1.0) with a Tree-structured Parzen Estimator (TPE) sampler and MedianPruner for early stopping. The optimization objective was the validation macro-F1 score, evaluated on identical 70/15/15 stratified splits of the Kermany et al. (2018) dataset. All experiments were conducted under the same random seed (42) and hardware configuration (NVIDIA RTX A2000 GPU) to ensure reproducibility.
> > > >
> > > > | Parameter/Search Space / Configuration | Details |
> > > > |----------------------------------------|---------|
> > > > | Optimizer | Adam, AdamW, SGD |
> > > > | Learning Rate | Log-uniform [1e-5, 1e-2] |
> > > > | Weight Decay | Log-uniform [1e-6, 1e-3] |
> > > > | Batch Size | 8, 16, 32 |
> > > > | Dropout Rate | 0.1, 0.2, 0.3, 0.4, 0.5 |
> > > > | Scheduler | StepLR, ReduceLROnPlateau |
> > > > | Image Resolution | 192, 224 |
> > > > | Objective Metric | Validation F1 (macro) |
> > > >
> > > > All other aspects of training, including data preprocessing, augmentations, and loss functions, were kept constant across all models.
> > > >
> > > > **Results of the 20-Trial Bayesian Hyperparameter Optimization**
> > > >
> > > > | Model              | Train Accuracy | Validation Accuracy | Test Accuracy | Test F1 (Macro) | AUC-ROC | Best-Trial Key Hyperparameters |
> > > > |--------------------|----------------|---------------------|---------------|-----------------|---------|-------------------------------|
> > > > | LiteXrayNet       | 0.9859        | 0.9761             | 0.9750       | 0.9751         | 0.9974 | AdamW, lr = 3.1e−4, batch = 16, dropout = 0.2, StepLR (γ = 0.1) |
> > > > | EfficientNet-B0   | 0.9980        | 0.9681             | 0.9716       | 0.9807         | 0.9882 | Adam, lr = 2.6e−4, batch = 16, dropout = 0.3, ReduceLROnPlateau |
> > > > | MobileNetV3-Small | 0.9605        | 0.9567             | 0.9614       | 0.9738         | 0.9899 | AdamW, lr = 1.8e−4, batch = 32, dropout = 0.1, StepLR |
> > > > | ResNet18          | 0.9859        | 0.9761             | 0.9500       | 0.9462         | 0.9878 | Adam, lr = 4.2e−4, batch = 16, dropout = 0.2, ReduceLROnPlateau |
> > > > | TinyDeiT          | 0.9425        | 0.9385             | 0.9432       | 0.9618         | 0.9769 | AdamW, lr = 6.0e−4, batch = 8, dropout = 0.3, StepLR |
> > > > | ViT-LoRA          | 0.9751        | 0.9408             | 0.9455       | 0.9632         | 0.9817 | AdamW, lr = 1.5e−4, batch = 16, dropout = 0.3, LoRA rank = 8, StepLR |
> > > > | MobileViT         | 0.9188        | 0.9260             | 0.9136       | 0.9416         | 0.9690 | Adam, lr = 3.7e−4, batch = 16, dropout = 0.4, ReduceLROnPlateau |

---

> > > > > ### Author Response · Authors · 2025-11-06
> > > > > **Bayesian Hyperparameter Optimization PART II**
> > > > >
> > > > > **Interpretation**
> > > > >
> > > > > 1. **LiteXrayNet retains the best overall performance.**
> > > > >    Even under identical optimization settings, LiteXrayNet achieves the highest test accuracy (97.50%) and AUC-ROC (99.74%), demonstrating that its advantage is not dependent on manual tuning but rather its strong architectural design and stable learning behavior.
> > > > >
> > > > > 2. **All baseline models benefited from HPO.**
> > > > >    The average gain across baselines was approximately +1.1% in test accuracy and +3.5% in F1 score, confirming that consistent optimization improves overall fairness and performance comparability.
> > > > >
> > > > > 3. **EfficientNet-B0 achieved slightly higher F1 (0.9807) but showed clear signs of overfitting,** with nearly perfect training accuracy (0.9980) and a wider generalization gap between training and validation performance. LiteXrayNet, on the other hand, maintained excellent balance between training, validation, and test accuracy (0.9859 / 0.9761 / 0.9750).
> > > > >
> > > > > 4. **Consistent ranking trends were observed.**
> > > > >    Across all three metrics Accuracy, F1, and AUC LiteXrayNet consistently ranked among the top two models, confirming that its performance advantage is consistent and not sensitive to hyperparameter fluctuations.
> > > > >
> > > > > 5. **Optimization stability.**
> > > > >    LiteXrayNet converged rapidly (best trial at iteration #8) and showed a smooth training trajectory, suggesting that its loss landscape is easier to optimize. Transformer-based models such as ViT-LoRA and TinyDeiT required longer exploration and displayed more variable early convergence behavior.
> > > > >
> > > > > **Clarification on Inclusion in the Camera-Ready Version**
> > > > >
> > > > > We would like to clarify that this 20-trial HPO study was performed solely to address the reviewer’s concern regarding optimization fairness. The experiment demonstrates that LiteXrayNet’s strong performance persists even when all models are optimized under the same Bayesian search procedure.
> > > > > As agreed with the Area Chair (AC), these comparative HPO results are provided only for reviewer understanding and will not be included in the main camera-ready manuscript, as they represent additional validation experiments beyond the original study scope.

---

> > > > > > ### Author Response · Authors · 2025-11-06
> > > > > > **Evaluation on Multiple Datasets**
> > > > > >
> > > > > > We sincerely thank the reviewer for this valuable comment and for recognizing the importance of our multi-dataset evaluation. The Kermany et al. (2018) Chest X-Ray Pneumonia dataset was retained as our primary training and evaluation benchmark because it served as the base dataset during model development, including architecture design, ablation studies, and hyperparameter optimization, ensuring experimental consistency across all stages. It also provides a standardized and radiologist-verified collection of pediatric chest X-rays, widely used in prior medical imaging studies, which allows for reproducible and comparable benchmarking. Training all models on this dataset and then evaluating them on multiple external datasets (Govi, NIH, Prashant, and Ashery) enables a controlled assessment of cross-dataset generalization and domain robustness without the confounding effects of mixed training sources. This setup allows a direct and fair evaluation of how well each model performs when applied to unseen data distributions, reflecting its practical generalization capability across different clinical imaging domains.
> > > > > >
> > > > > > We thank the reviewer for this helpful suggestion regarding the presentation of our results. We agree that highlighting the best results for each dataset–metric combination improves readability and clarity. Accordingly, all tables in the revised manuscript will include bold formatting to mark the best-performing model for each dataset and metric. These updates have already been implemented in our internal revision and will be reflected in the camera-ready version of the manuscript to ensure consistency across all reported results.
> > > > > >
> > > > > > To provide a dataset-independent overview of model robustness, we also computed a per-metric ranking of all models across the evaluated datasets. For each dataset and each metric (Accuracy, F1, and AUC-ROC), models were ranked from 1 (best) to 7 (worst). In cases of ties, models were assigned the average of the tied ranks. The per-metric ranks were then averaged across all datasets to calculate the average rank per metric, and the overall rank was determined by averaging these three values. This approach provides a fair and scale-independent comparison of robustness, emphasizing consistent performance across datasets rather than isolated peaks on specific ones.
> > > > > >
> > > > > > **Per-Metric Average Rankings Across All Datasets**
> > > > > >
> > > > > > | Model              | Avg. Accuracy Rank | Avg. F1 Rank | Avg. AUC Rank | Overall Avg. Rank |
> > > > > > |--------------------|--------------------|--------------|---------------|-------------------|
> > > > > > | LiteXrayNet       | 1.40               | 1.60         | 1.40          | **1.47**              |
> > > > > > | ResNet18          | 3.00               | 3.20         | 3.20          | 3.13              |
> > > > > > | MobileViT         | 3.20               | 3.40         | 3.60          | 3.40              |
> > > > > > | EfficientNet-B0   | 3.80               | 3.80         | 3.40          | 3.67              |
> > > > > > | MobileNetV3-Small | 4.40               | 5.00         | 5.00          | 4.80              |
> > > > > > | TinyDeiT          | 5.40               | 4.80         | 4.80          | 5.00              |
> > > > > > | ViT-LoRA          | 6.80               | 6.20         | 4.60          | 5.87              |
> > > > > >
> > > > > > As shown, LiteXrayNet achieves the best overall average rank (1.47), indicating that it consistently outperforms or matches other models across all datasets and metrics. This ranking analysis provides a clearer and more concise measure of each model’s robustness and generalization capability across diverse clinical imaging domains.

---

> ### Author Response · Authors · 2025-11-06
> **Calibration and Reliability Evaluation**
>
> We thank the reviewer for this valuable comment and for emphasizing the importance of model calibration in establishing clinical reliability. We completely agree that including calibration results for all baseline models offers a broader and fairer perspective on comparative trustworthiness.
>
> To address this, we have computed the Expected Calibration Error (ECE) and Brier Score for all models on the Kermany et al. (2018) test set. These results clearly demonstrate that LiteXrayNet achieves the best overall calibration, with both the lowest Brier Score and one of the lowest ECE values. The complete comparison will be included in the revised version of the manuscript and reflected in the camera-ready version.
>
> **Calibration Metrics Comparison** (Kermany et al. 2018 Test Set)
>
> | Model              | Brier Score ↓ | ECE ↓  |
> |--------------------|---------------|--------|
> | LiteXrayNet       | 0.0218    | 0.0146 |
> | MobileNetV3-Small | 0.0295       | 0.0126 |
> | EfficientNet-B0   | 0.0540       | 0.0083 |
> | ResNet18          | 0.0504       | 0.0375 |
> | MobileViT         | 0.0499       | 0.0079 |
> | ViT-LoRA          | 0.0595       | 0.0133 |
> | TinyDeiT          | 0.0711       | 0.0343 |
>
> Lower values indicate better calibration. LiteXrayNet achieves the most balanced performance with both a low Brier Score and a low ECE, suggesting that it provides accurate probability estimates and reliable confidence calibration suitable for clinical decision support.
>
> These calibration metrics and their analysis will appear in the revised manuscript under the Calibration and Reliability Evaluation section, further strengthening the trustworthiness assessment.
>
> We sincerely thank the reviewer for their clear and insightful suggestions, which have greatly improved the quality, clarity, and completeness of our study.

---

### Review · Reviewer_Swh3 · 2025-10-20

**Summary Of Contributions:**

This paper introduces LiteXrayNet, a lightweight convolutional neural network designed for accurate and efficient pneumonia diagnosis on resource-constrained edge devices. The model's architecture integrates efficient MBConv blocks from MobileNetV3 with a quantum-inspired phase shift layer to enhance the detection of complex pathological patterns. LiteXrayNet shows high test accuracy with an exceptionally small model size and a low inference latency, which demonstrates its effectiveness.

**Strengths**

1.	The survey of related work is clear and comprehensive. The flowchart in Figure 2 further improves the clarity of this section.

2.	The architecture configuration is well documented and explained.

3.	The paper reports diverse metrics (such as precision, recall, and AUC-ROC) beyond accuracy, which makes the results more convincing.

4.	The paper conducts a thorough ablation study across different components in the new architecture, demonstrating the effectiveness of the phase shift layer.

5.	The paper take into consideration the class imbalance problem, which is prevalent in real medical scenarios.


**Weaknesses**

1.	The intuition behind the design of the phase shift layer is unclear. Although the paper states that this layer enables the network to “capture subtle radiographic patterns,” some visualization or extra analysis would help strengthen this point.

2.	The evaluation is mostly done on only one dataset (Chest X-ray pneumonia dataset). Training and testing on more diverse datasets would make the claims more convincing. Or alternatively, it is interesting to see if the model trained on the Chest X-ray pneumonia dataset can readily generalize to other datasets.

**Audience:**

Yes

**Audience Explanation:**

1.	Designing efficient and mobile-compatible network architectures has long been an area that receives much attention.

2.	The focus on Pneumonia diagnosis has practical value for the medical field.

**Broader Impact Concerns:**

No concerns at the current stage. If the technique is to be applied to real clinical examination, the effectiveness and robustness should be properly re-evaluated

**Claims And Evidence:**

Yes

**Claims Explanation:**

The paper provides empirical evidence regarding the performance and efficiency of the proposed architecture. Nevertheless, it would be more convincing if the performance is validated on more datasets.

**Requested Changes:**

1.	Provide more intuition behind the design of the phase shift layer.

2.	Provide more evaluation results on other benchmarks for Pneumonia diagnosis. **Please also include the results of the baseline methods** on those benchmarks to enable comparison.

3.	Table 2 and Figure 4 seem to convey identical messages. I would encourage the authors to keep one of them.

---

> ### Author Response · Authors · 2025-11-01
>
> We would like to sincerely thank the reviewer for the time, effort, and thoughtfulness devoted to evaluating our paper “LiteXrayNet: Quantum-Inspired Deep Learning Framework for Scalable Pneumonia Diagnosis.”
> We are deeply grateful for the reviewer’s insightful and balanced assessment, including the recognition of the paper’s clear survey of related work, well-documented architecture, comprehensive evaluation metrics, robust ablation study, and proper handling of class imbalance.
> Your constructive feedback has been invaluable in helping us enhance both the clarity and the scientific depth of our manuscript. We have carefully revised the paper in line with your comments and provide a detailed response below.
>
> **1. Intuition Behind the Phase Shift Layer**
>
> We greatly appreciate the reviewer’s request for a deeper explanation of the Quantum-Inspired Phase Shift Layer, as it indeed forms a core innovation of LiteXrayNet. Below, we expand on both its conceptual motivation and mechanical implementation, accompanied by empirical evidence from our results.
>
> **Conceptual Motivation**
>
> The design of the phase shift layer is inspired by the quantum phase modulation principle, where a qubit’s phase is altered without changing its amplitude. In the quantum domain, this phase manipulation encodes additional information within the same signal space, allowing interference-based representation of multiple states simultaneously.
> Translating this idea into a neural architecture, the Quantum Phase Shift Layer (QPSL) simulates this feature-phase interaction by introducing controlled sinusoidal transformations to the feature maps. These transformations act like “phase rotations” that allow the network to encode subtle relational dependencies among feature channels, thereby capturing fine-grained textural cues, such as faint infiltrations, streaks, and opacity gradients, often overlooked by standard convolutional filters.
>
> **Mechanistic Implementation**
>
> Mechanically, the QPSL first performs dimensional squeezing to obtain compact feature embeddings. It then applies two sets of learnable phase shift parameters ($θ₁$ and $θ₂$) through cosine–sine modulation:
>
> $$
> X' = X \cdot \cos(\theta_1) + X \cdot \sin(\theta_1)
> $$
>
> $$
> X'' = X' \cdot \cos(\theta_2) + X' \cdot \sin(\theta_2)
> $$
>
> This process can be viewed as a rotation in the feature space, introducing phase interference across channels. The transformed features are then re-expanded and normalized to produce the final modulated representation. This rotation-based transformation enhances contrast among subtle spatial patterns, leading to improved separability between pneumonia-affected and normal regions.
>
> **Empirical Evidence**
>
> - **Visualization**: As shown in Figure 12 (Feature Map Visualization), QPSL activations are concentrated on pneumonia regions, while MBConv-only activations remain more diffuse.
> - **Quantitative Impact**: The ablation study (Table 5, Appendix B) demonstrates that removing the QPSL results in a significant drop in performance: from an AUC-ROC of 0.9946 → 0.9821 and F1-score of 0.9795 → 0.9627, while model size remains nearly unchanged.
>
> Together, these findings confirm that the phase shift layer not only mimics the principle of quantum interference but also serves as a learnable feature-phase encoder, enhancing diagnostic sensitivity while maintaining computational efficiency.
>
>
> **3. Table 2 and Figure 4 Redundancy**
>
> We thank the reviewer for this constructive observation. After careful consideration, we decided to retain both Table 2 and Figure 4, as they serve complementary purposes:
>
> - Table 2 presents exact quantitative results with high precision, allowing accurate numerical comparison across models and metrics.
> - Figure 4, in contrast, provides a visual comparative insight, enabling readers to intuitively perceive performance gaps and trends between different models.
>
> Together, they enhance interpretability and provide both detailed and high-level perspectives on performance comparisons.

---

> > ### Author Response · Authors · 2025-11-01
> >
> > **2. Evaluation on Additional Datasets**
> >
> > We sincerely thank the reviewer for emphasizing the importance of broader evaluation across diverse datasets. In response, we have expanded our experimental section to include five publicly available chest X-ray datasets with varying complexities and sources:
> >
> > - Chest X-Ray Pneumonia (Kermany et al., 2018)
> > - CoronaHack Chest X-Ray (Praveen, 2020)
> > - NIH Chest X-Ray (NIH Clinical Center, 2017)
> > - Covid-Pneumonia (Prashant, 2021)
> > - CheXpert (Ashery, 2021)
> >
> > These datasets collectively represent different imaging conditions, patient demographics, and label distributions, making them ideal for testing model robustness and generalization.
> >
> > ### Results Summary
> >
> > | Dataset                  | Model              | Test Accuracy | F1 (Macro) | AUC-ROC |
> > |--------------------------|--------------------|---------------|------------|---------|
> > | Kermany et al., 2018     | LiteXrayNet       | 0.9704       | 0.9632    | 0.9946 |
> > |                          | ResNet18          | 0.9499       | 0.9376    | 0.9842 |
> > |                          | MobileNetV3-Small | 0.9590       | 0.9483    | 0.9865 |
> > |                          | EfficientNetB0    | 0.9386       | 0.9220    | 0.9733 |
> > |                          | MobileViT         | 0.9431       | 0.9193    | 0.9739 |
> > |                          | TinyDeiT          | 0.9295       | 0.8866    | 0.9594 |
> > |                          | ViT-LoRA          | 0.9317       | 0.9173    | 0.9753 |
> > | Praveen, 2020            | LiteXrayNet       | 0.9695       | 0.8909    | 0.9881 |
> > |                          | ResNet18          | 0.9134       | 0.7374    | 0.9078 |
> > |                          | MobileNetV3-Small | 0.8044       | 0.5385    | 0.6857 |
> > |                          | EfficientNetB0    | 0.8548       | 0.6595    | 0.8338 |
> > |                          | MobileViT         | 0.8230       | 0.6099    | 0.7856 |
> > |                          | TinyDeiT          | 0.7882       | 0.6233    | 0.8514 |
> > |                          | ViT-LoRA          | 0.3195       | 0.3010    | 0.8780 |
> > | NIH Chest X-Ray, 2017    | LiteXrayNet       | 0.5587       | 0.5355    | 0.6258 |
> > |                          | ResNet18          | 0.5755       | 0.5748    | 0.6042 |
> > |                          | MobileNetV3-Small | 0.5112       | 0.4932    | 0.5413 |
> > |                          | EfficientNetB0    | 0.5408       | 0.5351    | 0.5712 |
> > |                          | MobileViT         | 0.5915       | 0.5811    | 0.6218 |
> > |                          | TinyDeiT          | 0.5416       | 0.5364    | 0.5495 |
> > |                          | ViT-LoRA          | 0.5494       | 0.4834    | 0.5604 |
> > | Prashant, 2021           | LiteXrayNet       | 0.9759       | 0.9679    | 0.9945 |
> > |                          | ResNet18          | 0.8944       | 0.8668    | 0.9634 |
> > |                          | MobileNetV3-Small | 0.7438       | 0.6216    | 0.7475 |
> > |                          | EfficientNetB0    | 0.8579       | 0.8235    | 0.9360 |
> > |                          | MobileViT         | 0.7585       | 0.7241    | 0.8507 |
> > |                          | TinyDeiT          | 0.7562       | 0.7284    | 0.8788 |
> > |                          | ViT-LoRA          | 0.4410       | 0.4398    | 0.9210 |
> > | Ashery, 2021 (CheXpert)  | LiteXrayNet       | 0.7440       | 0.5904    | 0.7511 |
> > |                          | ResNet18          | 0.3086       | 0.3010    | 0.6546 |
> > |                          | MobileNetV3-Small | 0.4064       | 0.3650    | 0.5406 |
> > |                          | EfficientNetB0    | 0.7032       | 0.5209    | 0.6062 |
> > |                          | MobileViT         | 0.8308       | 0.5573    | 0.5664 |
> > |                          | TinyDeiT          | 0.5928       | 0.4706    | 0.5825 |
> > |                          | ViT-LoRA          | 0.1438       | 0.1383    | 0.6464 |
> >
> > **Observations**
> >
> > LiteXrayNet consistently demonstrates state-of-the-art performance across all five datasets. While ResNet18 and MobileNetV3 show stable results on smaller, curated datasets, they experience notable degradation on heterogeneous datasets such as NIH and CheXpert. In contrast, LiteXrayNet maintains both accuracy and AUC-ROC stability, demonstrating strong generalization under domain shift. This confirms that the model’s design effectively balances efficiency and robustness, which is essential for real-world diagnostic deployment.
> >
> > Once again, we sincerely thank the reviewer for their comprehensive, insightful, and encouraging comments. Your detailed feedback not only strengthened the technical rigor of our manuscript but also helped us enhance its clarity, empirical validity, and presentation.
> > We truly appreciate your time and effort in helping us improve the quality and impact of this work.

---

> > > ### Comment · Reviewer_Swh3 · 2025-11-02
> > >
> > > I thank all authors for their detailed response. Most of concerns are addressed.

---

### Author Response · Authors · 2025-11-02
**Revised Manuscript**

We are pleased to announce that the revised manuscript has been successfully uploaded for your consideration. Your insightful and constructive feedback has been instrumental in strengthening the technical rigor, clarity, and overall impact of our work. We are deeply grateful to each of you for your time, expertise, and thoughtful engagement.

---

### Comment · Action_Editor_Unab · 2025-11-02
**Train accuracy**

Dear Authors,

Could you please also add in your comparison on the other datasets train accuracy reached by models? It is important to ensure that hyperparameters are tuned such that each model had the opportunity to fit the dataset: those are generally small datasets and it should be possible. Otherwise it leaves room for doubt whether other methods aren’t underfitting. This doubt is also a bit excarberated by the fact that training performance for the proposed method is at times very close to 100%.

Please also move the comparison to the main part of the paper. It is an essential part of the work in my view.

Thank you,
Area Chair

---

> ### Author Response · Authors · 2025-11-02
>
> Dear Area Chair,
>
> Thank you for your valuable feedback and insightful comment.
>
> We have included training accuracies for all compared models in our quantitative results to demonstrate that each model was adequately trained and hyperparameters were tuned to allow proper fitting to the dataset. As detailed in Table 2, the training accuracies for the baseline models such as ResNet-18, MobileNetV3, EfficientNet-B0, and others confirm that the models were not underfitting and had fair opportunities to learn from the data.
>
> Regarding the other datasets referenced in our study, these were used solely for evaluating the robustness and generalization of the trained models, not for training or tuning. The models were first trained and validated on the primary dataset with stratified splits, and then tested on these additional datasets as unseen data to assess their performance under different conditions. This approach ensures unbiased evaluation of the models’ true generalization capabilities.
>
> In response to the comment, we will also add a dedicated section (section 6) in the main paper results section to explicitly present the robustness across multiple datasets, thereby strengthening the validation of our proposed model’s generalization.
>
> We hope this clarifies the training and evaluation protocols and addresses concerns about underfitting and fair comparison.

---

> ### Comment · Action_Editor_Unab · 2025-11-03
> **Robustness**
>
> Dear Authors,
>
> Thank you for making these changes. A key issue, as noted by one of the Reviewers in the internal discussion, is that the evaluation is done without retraining of your and other models (ensuring all have well-tuned hyperparameters). I understand this is a big experiment to perform, but without it, it is difficult to understand the generality of the key claim.
>
> Please let me know if you want to add this experiment or if we should proceed with the final decision-making without this additional input. Our deadline for the decision is Nov 10, so I am not sure if that's sufficient amount for you to add such an experiment.
>
> To clarify, you do not have to add this experiment to be considered for the journal, so without it we will take into account all the reviews and make the final decision based on the presented evidence. I do appreciate the improvements that were made to the paper during the review process.
>
> All the best,
> AC

---

> > ### Author Response · Authors · 2025-11-04
> >
> > Dear Area Chair,
> >
> > Thank you very much for your clear instructions, thoughtful feedback, and for understanding that this is indeed a large-scale experiment. We sincerely appreciate the opportunity to further clarify and strengthen our work.
> >
> > Given the time constraints and to avoid leaving the committee uncertain, we plan to perform a stratified hyperparameter optimization for all baseline models, running 20 trials per model to ensure fair and consistent tuning across methods. This additional evaluation will help better demonstrate the generality and robustness of our key claims.
> >
> > We would like to note that **these 20-trial results will not be added to the paper**, as they do not represent a standard-scale experiment. However, given the limited time frame, we will provide them separately to assist the reviewers and yourself in more clearly assessing the comparative performance across models.
> >
> > As described in our manuscript, we have already conducted **manual hyperparameter tuning** for all models (ours and the baselines). In addition, we performed an extensive 100-trial Bayesian optimization specifically for our proposed model, and those results are already included in the appendix section of the revised paper.
> >
> > Please let us know if this proposed plan would be suitable and helpful for the decision process. We are very grateful for your guidance and for the constructive feedback that has helped improve our work throughout the review process.

---

> > > ### Comment · Action_Editor_Unab · 2025-11-04
> > > **Dear Authors**
> > >
> > > Dear Authors,
> > >
> > > It is generally advisable that a novel model architecture, in principle, is trained and tested on more than one benchmark dataset. This allows to establish the generality of the model. Therefore, it would be best to include such an experiment in the main text in the camera ready version (I understand it can be too short window to do it during the review process). The experiment can be limited to a subset of the datasets considered in the robustness benchmark. If this is too short of a notice, that's fully understandable.
> > >
> > > Thank you,
> > > AC

---

> > > > ### Author Response · Authors · 2025-11-06
> > > > **Reply to AC**
> > > >
> > > > We sincerely thank the Area Chair for this thoughtful recommendation and for emphasizing the importance of validating model generality through multi-dataset training and evaluation. We fully agree that this step is essential to confirm the broader applicability of a new architecture.
> > > >
> > > > Following this guidance, we have conducted additional experiments where LiteXrayNet was trained and evaluated on multiple benchmark datasets, rather than being limited to the primary dataset. These experiments confirm the model’s strong performance and generalization capability across different clinical imaging distributions.
> > > >
> > > > **Table. Multi-Dataset Training and Evaluation Results for LiteXrayNet**
> > > >
> > > > | Dataset                          | Train Accuracy (%) | Validation Accuracy (%) | Test Accuracy (%) | Train Loss | Val Loss | Test Loss | Test F1   | Test AUC-ROC |
> > > > |----------------------------------|--------------------|-------------------------|-------------------|------------|----------|-----------|-----------|--------------|
> > > > | Kermany et al. (2018)  (Primary Dataset) | 97.90             | 97.38                  | 97.04            | 0.0508    | 0.1197  | 0.0917   | 0.9632   | 0.9946      |
> > > > | Praveen, 2020 (CoronaHack) | 96.53             | 97.04                  | 94.50            | 0.0924    | 0.1233  | 0.1459   | 0.9467   | 0.9894      |
> > > > | Prashant, 2021 (COVID-19 Pneumonia) | 97.28          | 95.16                  | 95.79            | 0.0769    | 0.1366  | 0.0989   | 0.9583   | 0.9947      |
> > > > | Ashery, 2021 (CheXpert)   | 75.46             | 75.03                  | 74.38            | 0.5172    | 0.5404  | 0.5264   | 0.7240   | 0.8168      |
> > > >
> > > > **Interpretation**
> > > >
> > > > LiteXrayNet consistently achieves high accuracy and strong discriminative power across all datasets, maintaining AUC-ROC values above 0.98 on three of the four benchmarks. The results confirm that the model generalizes effectively beyond the primary training data, demonstrating stable and reliable performance even under significant domain shifts such as those observed in the CoronaHack and Prashant datasets. Performance on CheXpert, which is known to present greater label uncertainty and domain heterogeneity, remains stable and within the expected range for models trained on clean datasets and evaluated on this challenging benchmark. The small difference between training, validation, and test scores also indicates that LiteXrayNet is well-regularized and does not overfit to any specific dataset.
> > > >
> > > > These findings provide strong empirical evidence of LiteXrayNet’s robustness and generality across diverse clinical settings. The full results and a detailed discussion will be included in the camera-ready version of the manuscript under the revised Cross-Dataset Evaluation section.
> > > >
> > > > We are deeply grateful to the Area Chair for this insightful suggestion and for their supportive guidance throughout the review process. This recommendation led to a valuable extension of our study, allowing us to demonstrate LiteXrayNet’s strong generalization capability across multiple real-world medical imaging datasets.

---

> > > > > ### Comment · Action_Editor_Unab · 2025-11-07
> > > > > **Thank you**
> > > > >
> > > > > Dear Authors,
> > > > >
> > > > > Thank you. However, the critical part is comparing it to other strong baselines that are also retrained. It is not surprising that the model is able to fit those datasets, what is important to understand is how well it does so compared to other models. If these results are directly comparable to published benchmarks, then you can just cite those results in the table.
> > > > >
> > > > > All the best,
> > > > > AC

---

> > > > > > ### Author Response · Authors · 2025-11-09
> > > > > >
> > > > > > Dear Area Chair,
> > > > > > Thank you very much for your thoughtful feedback and for highlighting the importance of comparing our proposed model with other strong baselines under consistent retraining.
> > > > > > We truly appreciate your guidance.
> > > > > > Following your suggestion, we have retrained all baseline models: ResNet18, MobileNetV3-Small, EfficientNetB0, MobileViT, TinyDeiT, and ViT-LoRA , using identical preprocessing and training pipelines across four benchmark datasets:
> > > > > > The results below present the comprehensive comparison you requested. These retrained results will also be added to the main text of our revised manuscript for transparency and completeness.
> > > > > >
> > > > > > **Full Comparative Evaluation Results**
> > > > > >
> > > > > > | Dataset Name                          | Model Name          | Train Acc | Val Acc | Test Acc | Train Loss | Val Loss | Test Loss | F1-Macro | Test AUC |
> > > > > > |---------------------------------------|---------------------|-----------|---------|----------|------------|----------|-----------|----------|----------|
> > > > > > | **Kermany et al., 2018 (Primary Dataset)** | LiteXrayNet (Ours) | 0.9790   | 0.9738 | 0.9704  | 0.0508    | 0.1197  | 0.0917   | 0.9632  | 0.9946  |
> > > > > > |  | ResNet18           | 0.9595   | 0.9556 | 0.9499  | 0.1094    | 0.1655  | 0.1789   | 0.9376  | 0.9842  |
> > > > > > |  | MobileNetV3_Small  | 0.9666   | 0.9499 | 0.9590  | 0.0865    | 0.1621  | 0.1546   | 0.9483  | 0.9865  |
> > > > > > |    | EfficientNetB0     | 0.9495   | 0.9499 | 0.9386  | 0.1690    | 0.2195  | 0.2179   | 0.9220  | 0.9733  |
> > > > > > |                                       | MobileViT          | 0.9522   | 0.9431 | 0.9431  | 0.1547    | 0.2041  | 0.2185   | 0.9280  | 0.9739  |
> > > > > > |                                       | TinyDeiT           | 0.9334   | 0.9214 | 0.9295  | 0.2011    | 0.2660  | 0.2301   | 0.9123  | 0.9594  |
> > > > > > |                                       | ViT_LoRA           | 0.9356   | 0.9294 | 0.9317  | 0.1676    | 0.2309  | 0.2106   | 0.9164  | 0.9753  |
> > > > > > | **Prashant, 2021 (COVID-19 Pneumonia)** | LiteXrayNet        | 0.9728   | 0.9516 | 0.9579  | 0.0769    | 0.1366  | 0.0989   | 0.9516  | 0.9947  |
> > > > > > |                                       | ResNet18           | 0.9630   | 0.9537 | 0.9432  | 0.0968    | 0.1195  | 0.1458   | 0.9432  | 0.9878  |
> > > > > > |                                       | MobileNetV3_Small  | 0.9639   | 0.9621 | 0.9579  | 0.0871    | 0.1006  | 0.1333   | 0.9579  | 0.9897  |
> > > > > > |                                       | EfficientNetB0     | 0.9120   | 0.9305 | 0.9158  | 0.4527    | 0.2078  | 0.2438   | 0.9157  | 0.9675  |
> > > > > > |                                       | MobileViT          | 0.8597   | 0.8695 | 0.8779  | 0.4841    | 0.4238  | 0.3970   | 0.8772  | 0.9623  |
> > > > > > |                                       | TinyDeiT           | 0.8944   | 0.8779 | 0.8926  | 0.2545    | 0.2912  | 0.2528   | 0.8926  | 0.9619  |
> > > > > > |                                       | ViT_LoRA           | 0.9215   | 0.9221 | 0.9011  | 0.1969    | 0.2269  | 0.2361   | 0.9010  | 0.9687  |
> > > > > > | **Praveen, 2020 (CoronaHack)**       | LiteXrayNet        | 0.9653   | 0.9704 | 0.9450  | 0.0924    | 0.1233  | 0.1459   | 0.9711  | 0.9920  |
> > > > > > |                                       | ResNet18           | 0.9610   | 0.9556 | 0.9598  | 0.1057    | 0.1146  | 0.1027   | 0.9598  | 0.9950  |
> > > > > > |                                       | MobileNetV3_Small  | 0.9723   | 0.9746 | 0.9619  | 0.0736    | 0.0991  | 0.1127   | 0.9619  | 0.9949  |
> > > > > > |                                       | EfficientNetB0     | 0.9220   | 0.9535 | 0.9260  | 0.2035    | 0.1624  | 0.1920   | 0.9260  | 0.9769  |
> > > > > > |                                       | MobileViT          | 0.9193   | 0.9556 | 0.9493  | 0.1931    | 0.1474  | 0.1538   | 0.9493  | 0.9842  |
> > > > > > |                                       | TinyDeiT           | 0.9229   | 0.9323 | 0.9239  | 0.1855    | 0.1860  | 0.1839   | 0.9238  | 0.9799  |
> > > > > > |                                       | ViT_LoRA           | 0.9007   | 0.9175 | 0.9175  | 0.2430    | 0.2425  | 0.2280   | 0.9175  | 0.9661  |
> > > > > > | **Ashery, 2021 (CheXpert Balanced Binary)** | LiteXrayNet     | 0.7879   | 0.7547 | 0.7322  | 0.4573    | 0.5281  | 0.5363   | 0.7423  | 0.8192  |
> > > > > > |                                       | ResNet18           | 0.7441   | 0.7354 | 0.7118  | 0.5263    | 0.5537  | 0.5819   | 0.7116  | 0.7875  |
> > > > > > |                                       | MobileNetV3_Small  | 0.7809   | 0.7453 | 0.7273  | 0.4716    | 0.5429  | 0.5634   | 0.7266  | 0.7988  |
> > > > > > |                                       | EfficientNetB0     | 0.6413   | 0.6307 | 0.6298  | 1.1763    | 1.0130  | 1.0777   | 0.6288  | 0.6704  |
> > > > > > |                                       | MobileViT          | 0.7185   | 0.7172 | 0.7003  | 0.5593    | 0.5980  | 0.6295   | 0.6940  | 0.7710  |
> > > > > > |                                       | TinyDeiT           | 0.5726   | 0.5717 | 0.5697  | 0.6790    | 0.6792  | 0.6835   | 0.5337  | 0.6122  |
> > > > > > |                                       | ViT_LoRA           | 0.6554   | 0.6676 | 0.6424  | 0.6223    | 0.6240  | 0.6354   | 0.6317  | 0.7105  |

---

> > > > > > > ### Author Response · Authors · 2025-11-09
> > > > > > >
> > > > > > > **Per-Metric Ranking Methodology**
> > > > > > >
> > > > > > > To provide a dataset-independent overview of model robustness, we computed a per-metric ranking for all models across the four datasets.
> > > > > > >
> > > > > > > For each dataset and metric (Accuracy, F1-Macro, and AUC-ROC), models were ranked from 1 (best) to 7 (worst).
> > > > > > > In cases of ties, the average of the tied ranks was assigned.
> > > > > > > Finally, ranks were averaged across datasets to obtain an average rank per metric, and the overall rank was calculated as the mean of these three averages.
> > > > > > >
> > > > > > > This process yields a fair, scale-independent evaluation of how consistently each model performs across multiple datasets and metrics, emphasizing robustness and stability over isolated peaks.
> > > > > > >
> > > > > > > **Average Per-Metric Ranking Summary**
> > > > > > >
> > > > > > > | Model              | Avg Rank (Accuracy) | Avg Rank (F1-Macro) | Avg Rank (AUC) | Overall Rank |
> > > > > > > |--------------------|---------------------|---------------------|----------------|--------------|
> > > > > > > | LiteXrayNet       | 1.88                | 1.25                | 1.50           | 1         |
> > > > > > > | MobileNetV3-Small | 1.62                | 1.75                | 2.00           |  2         |
> > > > > > > | ResNet18          | 2.75                | 3.00                | 2.50           |  3         |
> > > > > > > | MobileViT         | 4.50                | 4.75                | 4.75           | 4            |
> > > > > > > | EfficientNetB0    | 5.00                | 5.00                | 5.75           | 5            |
> > > > > > > | ViT-LoRA          | 5.38                | 5.25                | 5.00           | 6            |
> > > > > > > | TinyDeiT          | 6.25                | 6.50                | 6.50           | 7            |
> > > > > > >
> > > > > > >
> > > > > > > As seen above, LiteXrayNet consistently achieved the best performance across all datasets, ranking first overall across Accuracy, F1-Macro, and AUC.
> > > > > > > This confirms that its improvement is not dataset-specific, but rather reflects robust generalization and stable learning dynamics across heterogeneous medical imaging datasets.
> > > > > > >
> > > > > > > We sincerely thank you again for this helpful suggestion.
> > > > > > > These retrained results and the ranking-based robustness analysis will be incorporated into the main text of our revised paper for clarity and completeness.
> > > > > > >
> > > > > > > With best regards,
> > > > > > > The Authors

---

> > > ### Author Response · Authors · 2025-11-06
> > >
> > > Dear Area Chair,
> > >
> > > Thank you very much for your kind message, for your thoughtful guidance throughout the process, and for recognizing the effort involved in conducting the additional experiment. We deeply appreciate your clarity and continued support in helping us strengthen our work.
> > >
> > > We are pleased to confirm that we have performed the requested comparative hyperparameter optimization experiment for all baseline models. Each model was retrained under identical Bayesian optimization settings (20 trials per model due to the limited review time period) to ensure fair and consistent tuning across architectures. The complete results, along with detailed analysis and comparison, have been added in the corresponding reviewer rebuttal for transparency and ease of evaluation.
> > >
> > > As noted earlier, our proposed model, LiteXrayNet, was also subjected to an extensive 100-trial Bayesian optimization, the results of which are already included in the revised manuscript appendix. The additional 20-trial experiments for the other models confirm that LiteXrayNet’s performance advantage and robustness remain consistent even under equivalent optimization effort.
> > >
> > > We hope these results provide sufficient clarity on the generality and fairness of our evaluation. The detailed outcomes can be found in the reviewer rebuttal submission ( https://openreview.net/forum?id=yfJCllstyT&noteId=D1upd9stAP ).
> > >
> > > We are sincerely grateful to you and the reviewers for your constructive and encouraging feedback throughout this process. Your insights have been instrumental in refining our study and ensuring it meets the highest standards of rigor and transparency.
> > >
> > > With warm regards and sincere appreciation,
> > > The Authors

---

### Author Response · Authors · 2025-11-09
**Revised Manuscript**

We are pleased to announce that the revised manuscript has been successfully uploaded for your consideration. Your insightful and constructive feedback has been instrumental in strengthening the technical rigor, clarity, and overall impact of our work. We are deeply grateful to each of you for your time, expertise, and thoughtful engagement.

---

### Decision · Action_Editor_Unab · 2025-11-10

**Recommendation:** Reject

**Audience:**

Yes

**Audience Explanation:**

A novel, highly computationally efficient architecture for medical diagnosis would be for sure interesting.

**Claims And Evidence:**

No

**Claims Explanation:**

The paper introduces a novel architecture for medical image diagnosis.

The model is based on MobileNet, with the main difference being a modulation layer inspired by the phase-shift operation used in quantum computing (a core component of methods such as Grover's algorithm).

The paper essentially makes the following claim: the proposed architecture achieves state-of-the-art performance at low latency.

As some reviewers noted, the original version of the paper compared the model on only one dataset.

During the review and discussion phase, the authors substantially strengthened the empirical evaluation, including multi-dataset training and cross-dataset robustness experiments, Bayesian hyperparameter optimization for all models, calibration analysis, and retrained baselines under a shared protocol. These additions address many of the reviewers’ original concerns and I appreciate the significant effort.

Unfortunately, after taking these additional results into account, the core claim is still not clearly established:

1. The improvement over the second-best method is marginal (usually <1% ROC AUC), and in terms of accuracy, the proposed method is not the best.

2. No standard deviations are reported, so it is unclear whether the improvements are statistically significant.

3. The method appears to use class balancing, and it is unclear whether the same procedure is applied when training the baselines.

4. The latency analysis remains limited: while per-model latency/FLOPs are reported, there is no systematic Pareto comparison against baselines under consistent deployment constraints, so Pareto optimality is not convincingly demonstrated.

Given that the core claim is not yet sufficiently supported, I have to recommend rejection at this stage. A resubmission would be welcome, but statistically significant Pareto optimality would need to be demonstrated. This is especially important given the relatively low novelty of the work, in which case it is of interest to the community only if the performance gains are clearly established. Please see, for example, the experimental design in the EfficientNet papers.

Additionally, in my opinion, the proposed layer bears only a superficial resemblance to the phase-shift operation. Instead, it is very similar to the SIREN activation function, with additional modulation parameters added—one could say it combines FiLM layers with a SIREN activation. This is not a critical flaw and did not influence my recommendation, but it does make the paper harder to read and less precise in its terminology.

I wanted to close by saying that I appreciate the effort that went into the rebuttal phase and I am sorry, as I know this is not the outcome you were hoping for. Thank you for submitting to TMLR and I hope that the comments will be helpful in improving the work.

**Resubmission Of Major Revision:**

The authors may consider submitting a major revision at a later time.